# MEM1: Learning to Synergize Memory and Reasoning for Efficient Long-Horizon Agents

**Zijian Zhou**[12*], **Ao Qu**[13*], **Zhaoxuan Wu**[1], **Sunghwan Kim**[4], **Alok Prakash**[1]
**Daniela Rus**[13], **Bryan Kian Hsiang Low**[12], **Paul Pu Liang**[13]

[1]Singapore-MIT Alliance for Research and Technology Centre, Singapore
[2]Department of Computer Science, National University of Singapore, Singapore
[3]Massachusetts Institute of Technology, USA
[4]Yonsei University, South Korea

`{zhou_zijian,wu.zhaoxuan}@u.nus.edu, lowkh@comp.nus.edu.sg`
`{qua,ppliang}@mit.edu, alok.prakash@smart.mit.edu, rus@csail.mit.edu`
`kimsh8564@yonsei.ac.kr`

## Abstract

Modern language agents often need to solve long-horizon tasks requiring multiple turns of interactions with the environment, where they retrieve external information, adapt to observations, and answer interdependent queries. Yet, most LLM systems rely on full-context prompting, appending all past turns regardless of their relevance. This leads to unbounded memory growth, increased computational costs, and degraded reasoning performance on out-of-distribution input lengths due to LLM forgetting the context. We introduce **MEM1**, an end-to-end reinforcement learning framework that enables agents to operate with nearly **near constant context size** when solving long-horizon tasks. At each turn, MEM1 updates a **compact shared internal state** that jointly supports memory consolidation and reasoning. Leveraging reinforcement learning (RL) and rollout trajectory truncation, we train a MEM1 agent to develop internal states that integrate prior memory with new observations from the environment while strategically discarding irrelevant or redundant information. Experiments across three domains, including internal retrieval QA, open-domain web QA, and multi-turn web shopping, show that MEM1-7B improves performance by $3.5\times$ while reducing memory usage by $3.7\times$ compared to Qwen2.5-14B-Instruct on an augmented multi-hop QA dataset with 16 **objectives** in each task, and **generalizes beyond the training horizon**. Our results demonstrate the promise of reasoning-driven memory consolidation as a scalable alternative to existing solutions for training long-horizon task-solving agents with multiple interactions, where both efficiency and performance are optimized. Code can be found at https://github.com/MIT-MI/MEM1.

## 1 Introduction

Large language models (LLMs) have shown remarkable performance in single-turn tasks such as question answering, summarization, and code generation (Brown et al., 2020; Touvron et al., 2023). However, emerging real-world applications increasingly operate over multiple turns—searching documents, interacting with environments (Zhou et al., 2024), and making decisions based on evolving external information (Wang et al., 2024). Examples include AI search agents such as Perplexity AI (Perplexity AI, 2024) that automate complex tasks by iteratively gathering information, and web-navigation agents such as OpenManus (OpenManus, 2025) and BrowserUse (Müller & Žunič, 2024), which must complete goals across dozens of interactive turns.

Unlike traditional tasks where the input is static or self-contained, long-horizon settings often involve answering a sequence of related questions, requiring the agent to continuously retrieve new

---

* Equal contribution. Correspondence: `zhou_zijian@u.nus.edu, qua@mit.edu`

information, revise beliefs, and adapt to evolving contexts over time. For instance, consider a research assistant tasked with "What's the evidence for X?". Subsequent queries like "Who published it?" require further information retrieval, while "Is the source credible?" calls for self-reflection and assessment. Each query builds on the previously collected and accumulated information. Similarly, a shopping assistant may be first asked "Which product is cheapest?", then "What are its reviews?", and "Is it compatible with my device?". These interactions span multiple turns, featuring evolving contexts and compound reasoning.

In long-horizon systems, a common strategy is to append all past observations, actions, and thoughts to the context at each step (Wei et al., 2022; Yao et al., 2023). This creates three challenges. **(1) Growing inference cost and memory usage**. Transformer-based LLMs scale with $O(N^2)$ compute (or $O(N)$ with KV caching) and $O(N)$ memory as context length $N$ grows (Vaswani et al., 2017), forcing deployments to reserve large GPU memory and often wasting resources (Kwon et al., 2023; Zheng et al., 2024). **(2) Generalization limits beyond the training horizons**. Contexts longer than those seen during training push the model out-of-distribution, reducing its ability to reason reliably (Yoon et al., 2024). **(3) Overloaded context and forgetting**. Redundant or irrelevant content that grows with agentic interaction dilutes attention and makes the model prone to forgetting important details, even when they remain technically available in the context (An et al., 2025; Liu et al., 2024; Wu et al., 2025).

Recent progress in long-context modeling largely targets static inputs (*e.g.,* long documents) and does not address multi-turn interaction with external environments (Beltagy et al., 2020; Gu et al., 2025). Some other approaches introduce external memory modules (*e.g.,* summarizers or retrievers) (Yoon et al., 2024; Li et al., 2023; Chhikara et al., 2025; Xu et al., 2025), but these are typically trained separately and cannot be optimized end-to-end with the agent's policy. This also introduces additional engineering overhead, as engineers must manage and integrate two separate models. Meanwhile, existing works on tool-using agent systems trained with reinforcement learning leave memory management unsolved, letting the prompt length grow unboundedly (Jin et al., 2025; Zheng et al., 2025). A natural question is raised: ***Can a language model learn to consolidate its memory as part of its reasoning process*** *so that it retains only what is essential for solving the task?*

Motivated by this question, we present MEM1: **M**emory-**E**fficient **M**echanism via learning **1**-step integrated reasoning and consolidation—a method for training LLM agents that maintain nearly constant memory usage across arbitrarily long horizons. At each turn, the model updates a consolidated state composed of prior memory and newly obtained information. This consolidated state becomes the agent's only retained memory, allowing all observations obtained via external tool use to be discarded after use, which prevents prompt expansion altogether (illustrated later in Sec. 4.2). A key insight of our method is that inference-time reasoning (Wei et al., 2022; DeepSeek-AI et al., 2025; Muennighoff et al., 2025; Yue et al., 2025) serves two purposes: While reasoning about the current query, the model also extracts and stores the essential information it needs for the future. By unifying reasoning and memory consolidation, MEM1 enables the agent to both reason and remember within a shared representational space, without requiring extra modules or architectural changes. We train this behavior end-to-end with reinforcement learning (RL) (Sutton & Barto, 2018; Zhu et al., 2023), optimizing for task success via verifiable rewards (Shao et al., 2024). Although not explicitly optimized for memory efficiency through reward signals, the agent learns to manage memory as part of its policy, resulting in near-constant memory usage across long horizons.

We empirically evaluate MEM1 on (1) a multi-turn information retrieval task, where we augment standard single-objective QA datasets into multi-objective settings by composing $N$ multi-hop questions, and (ii) the WebShop environment (Yao et al., 2022), which requires the agent to perform multiple steps of interaction with a textual web environment. Across these diverse settings, MEM1 consistently matches or exceeds the performance of leading baselines while achieving efficiency gains of up to $3.5\times$ in memory usage. Moreover, agents trained on our 2-objective augmented tasks generalize robustly to much harder cases with up to 16 sequential objectives. At this extreme, MEM1 not only outperforms all baselines in accuracy but also reduces peak memory usage by $1.27\times$ and accelerates inference by $1.78\times$ relative to the strongest uncollapsed baseline.

## 2 RELATED WORK

**LLM agents in multi-turn environment.** LLM-based agents have evolved from handling single-turn queries to serving as autonomous agents capable of multi-turn interactions such as web navigation (Yao et al., 2022; Zhou et al., 2024) and complex research (Zheng et al., 2025). To enable such capabilities, Yao et al. (Yao et al., 2023) introduced the ReAct (*i.e.,* Reason + Act) framework, which enhances LLMs' ability to interact with external environments by interleaving reasoning and action. Building on this reasoning-acting prompting paradigm, subsequent works have explored ways to improve agent performance through natural language feedback, enabling iterative refinement (Shinn et al., 2023; Madaan et al., 2023). Recently, inference-time scaling has emerged as a promising direction for enabling complex reasoning, with prior research incorporating evaluators (*e.g.,* verifier, reward model) (Snell et al., 2024; Liu et al., 2025) or world models (Chae et al., 2024). In addition, there are two major lines of training approaches: (1) behavior cloning (BC), which involves imitating expert trajectories to guide agent behavior by supervised fine-tuning (SFT) (Yin et al., 2023; Deng et al., 2023; Cheng et al., 2024), and (2) reinforcement learning (RL), which optimizes agent policies by incentivizing desirable outcomes through rewards (Song et al., 2024; Bai et al., 2024; Qi et al., 2024). These methods aim to align the agents' behaviors with task objectives, enabling more robust and generalizable performance.

**Context management for LLM agents.** A widely adopted approach to context management in LLM-based agent systems involves appending all prior information, such as observations, intermediate thoughts, and actions, into the prompt at each interaction turn (Yao et al., 2023). While this method is straightforward and effective when the number of interactions required is small, it results in unbounded context growth, leading to linearly scaled inference memory. Moreover, long contexts often contain irrelevant or redundant information, which impairs the model's reasoning capabilities (An et al., 2025; Liu et al., 2024; Wu et al., 2025). To mitigate these issues, recent studies have proposed external memory frameworks, including retrieval-augmented generation and summarization modules (Yoon et al., 2024; Li et al., 2023; Chhikara et al., 2025; Xu et al., 2025) and hierarchical working memories (Hu et al., 2025b; Rasmussen et al., 2025; Sun & Zeng, 2025). However, these methods are typically applied independently of the agent's policy, creating a disconnect between memory and the reasoning process. In addition, managing and integrating such modules often incurs extra computational overhead and system complexity. Despite these advancements, many RL approaches for training LLM agents still rely on accumulating the full interaction history as memory (Jin et al., 2025; Zheng et al., 2025; Qi et al., 2024), leaving memory management during training an underexplored area. In this work, we seek to bridge this gap by tightly integrating memory with the agent's reasoning process, thereby enabling more efficient and context-aware decision-making.

## 3 MEM1

Complex reasoning tasks often require an iterative process of information gathering and synthesis, as seen in applications such as deep search (Perplexity AI, 2024) and web-based agents (Nakano et al., 2022; Gur et al., 2024). We consider an interactive agent operating in a multi-turn environment with vocabulary space $\mathcal{V}$. The generation process of an agent is defined as a Markov Decision Process (MDP) parameterized by $(\mathcal{S}, \mathcal{A}, \pi, r)$, where $\mathcal{A}$ represents the action space, $\mathcal{S}$ represents the state space, $\pi : \mathcal{S} \times \mathcal{A} \times \mathcal{S} \to [0, 1]$ is the transition distribution (i.e., the policy), and $r : \mathcal{S} \to \mathbb{R}$ is the reward function. A trajectory generated by a policy $\pi$ is denoted with $\tau := \{(a_t, s_t)|t \in \{1, 2, \ldots, |\tau|\}\}$. At each step $t$, the agent receives an observation $O_t \in \mathcal{A}^+$ (from external tools, APIs, or the environment), maintains an internal state $S_t \in \mathcal{A}^+$ (reasoning history and memory), and produces an action $A_t \in \mathcal{A}^+$ (e.g., answering a question or issuing a query), where $\mathcal{A}^+$ represents the set of sequences of $a \in \mathcal{A}$. The agent's goal is to maximize task success across long-horizon trajectories while keeping the retained context bounded. Formally, let $r : \mathcal{V} \to \mathbb{R}$ be the reward function. The learning problem is a maximization over the model parameters $\theta$

$$\text{argmax}_\theta \, \mathbb{E}_{Q \in \mathcal{Q}, \tau \sim \pi_{\theta, Q}} \left[ \sum_{(a_t, s_t) \in \tau} r(s_t) \right],$$

where $\tau = (S_i, A_i, O_i)_{i=1}^{n-1} \cap (S_n, A_n)$ is a trajectory sampled from policy $\pi_{\theta, Q}$ with $n$ turns and $\mathcal{Q}$ refers to the set of questions. In this work, we primarily consider tasks with verifiable rewards (i.e., $r$ is a rule-based mapping). A long, multi-turn reasoning task is characterized a large $n$, requiring the agent to iteratively perform a long series of searches and reasoning to derive the answer $A_n$.

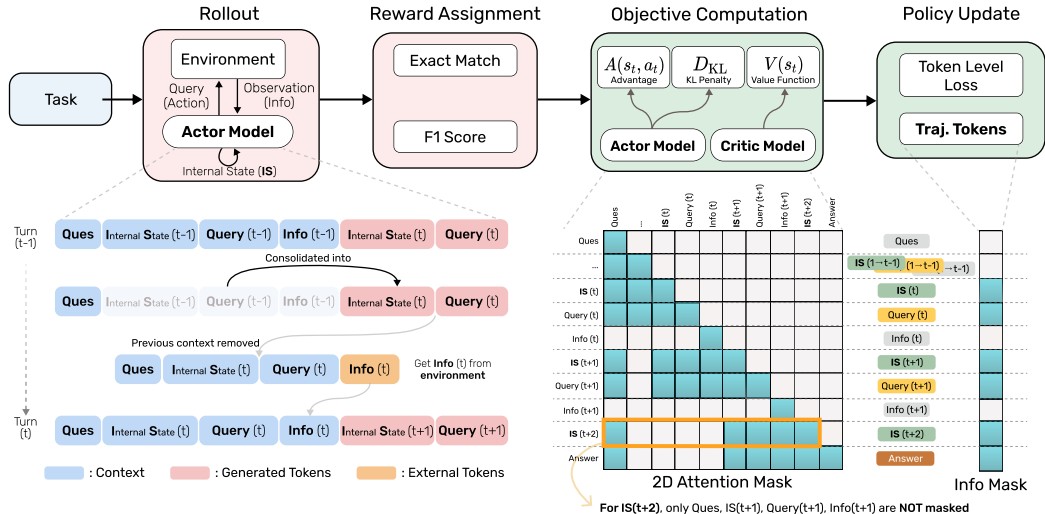

Figure 1: (Top): the RL pipeline used to train MEM1. (Bottom left): The evolution of context in MEM1–old internal states ($S$), query/answer ($A$), and external information ($O$) are cleared as new states enter the context during rollout. (Bottom right): the 2D attention mask used during the objective computation stage. The mask is applied during the forward pass to compute action log-probabilities for the actor model and state value estimates for the critic model. During policy update, an information mask is then applied to mask out $O$ tokens not generated by the model itself.

### 3.1 MEMORY AS PART OF REASONING

To achieve a constant memory, MEM1 is particularly trained to iteratively refine its understanding by processing new information in conjunction with a consolidation of its prior state. At each turn $i$, the agent produces a new $S_i$, which summarizes past information and reasons about subsequent actions. Following this, the agent generates an action $A_t$—a subsequent query or the answer if a direct response is warranted. If the agent issues a query, the corresponding feedback from the environment $O_i$ is appended to the trajectory. At the next turn, $i + 1$, the agent consolidates the tuple $(S_i, A_i, O_i)$ into a new $S_{i+1}$, which serves as the basis for further interactions. After each turn, $(S_i, A_i, O_i)$ is pruned from the context, effectively compressing memory and preventing prompt bloat. Fig. 1 (bottom left) illustrates the evolution of the model's context over time. At each turn, the agent retains at most two $S$'s, two $A$'s, and one $O$, ensuring bounded and efficient memory usage. The detailed rollout algorithm is in Alg. 1 of App. B.5.

RL offers a powerful mechanism for shaping agent behavior through reward signals (Sutton et al., 2000). In MEM1, we leverage this framework to incentivize effective state consolidation by designing environments in which the agent is rewarded only when it strategically retains and integrates useful information. Specifically, we construct tasks that require numerous interactions with the environment to arrive at a correct answer (see Sec. 3.3). Success depends on the agent's ability to rely on information collected along the inference path. At each turn, we prune the agent's context to retain only the most recent internal state $S$, forcing the agent to perform memory consolidation as part of its reasoning process. Without access to full historical context, the agent must learn to preserve and update relevant knowledge internally in order to reap the reward. This learning procedure mirrors how humans cultivate memorization skills through structured tasks such as Sudoku or crosswords (A Cognitive Connection, 2024), where success hinges on selectively attending to key information and building upon it. Over time, such tasks help individuals develop cognitive strategies that jointly support efficient memorization and reasoning, similar to our RL method for training MEM1.

### 3.2 MASKED TRAJECTORY FOR POLICY OPTIMIZATION

Popular RL algorithms (Schulman et al., 2017; Hu et al., 2025a) update the policy with policy gradient, which requires the calculation of $\nabla_\theta \pi_{\theta,Q}(a_t, s_t)$. For LLM, $\pi_{\theta,Q}(a_t, s_t)$ is viewed as the logit of the output $a_t$ of the model, where the input is $s_t$. Existing RL frameworks typically

compute the $\nabla_\theta \pi_{\theta,Q}(a_t, s_t)$'s for all pairs $(a_t, s_t) \in \tau$ by passing the entire rollout trajectory $\tau$ through the LLM once (i.e., prefilling). However, as MEM1 dynamically consolidates its context, the tokens $a_t$ do not belong to one single trajectory $\tau$. A naive solution is to break each turn into a sub-trajectory $\tau_i = (S_i, A_i, O_i)$, where $i$ represents the $i$th interaction turn. However, this approach introduces difficulties (at least implementation-wise) in computing the temporal difference $\delta_t = r(s_t) + V(s_{t+1}) - V(s_t)$ for the last token in the current sub-trajectory $\tau_i$, as $V(s_{t+1})$ is calculated in a separate sub-trajectory $\tau_j, j \neq i$. Here $V : \mathcal{S} \to \mathbb{R}$ is the value function.

To overcome this challenge, we introduce a masked trajectory that compresses $\{\tau_1, \tau_2, \ldots, \tau_n\}$ for a task with $n$ turns into a consolidated full trajectory $\tau_{full} = (\tau_1, \tau_2, \ldots, \tau_n) = (S_1, A_1, O_1, S_2, A_2, O_2, \ldots, S_n, A_n)$. The full trajectory encodes all information needed for accurate policy learning while respecting MEM1's memory consolidation at each turn. Note that $\tau_{full}$ is a "stitched" trajectory where the $S_i$'s and $A_i$'s do not belong to the same roll-out. As such, to ensure that policy gradients $\nabla_\theta \pi_{\theta,Q}(a_t, s_t)$ are correctly computed under this consolidated memory regime, we apply a **two-dimensional attention mask** (S., 2024) across $\tau_{full}$. This mask restricts each token's attention to only the tokens retained in memory at the time that token was generated. Specifically, let the attention mask for the $t$th token in the $i$th turn be $\text{Attn}_t = \mathbf{1}_{a \in \{S_{i-1}, A_{i-1}, O_{i-1}, S_i, A_i, O_i\}} \times \mathbf{1}_{a \in \{a_k | k \in \{1,2,\ldots,t\}\}}$. We can compute the policy for the $i$th turn $\pi_{\theta,Q,\tau_i}(a_t, s_t) = \pi_{\theta,Q,\tau_{full}}(a_t, s_t \times \text{Attn}_t)$ and $\nabla_\theta \pi_{\tau_i}$ accordingly. Algorithmically, this can be achieved by first constructing the attention mask $\text{Mask} = (\text{Attn}_1, \ldots, \text{Attn}_T)$ and masking the attention matrix during the transformer forward. Fig. 1 (bottom right) shows the masking mechanism that enables stable and accurate policy optimization under MEM1's memory-constrained execution.

## 3.3 MULTI-OBJECTIVE TASK DESIGN

Although MEM1 is designed to address the tasks involving multi-turn interaction with the external world, there are limited publicly available datasets that support training for such long-horizon interactive processes. Existing benchmarks, such as HotpotQA (Yang et al., 2018), Bamboogle (Press et al., 2023), and 2wiki (Ho et al., 2020), are often cited as multi-turn benchmarks, yet they typically involve only two information-seeking steps. Moreover, these datasets are not explicitly structured to support long-horizon interactions that necessitate the agent to manage the memory state.

To bridge this gap, we introduce a novel task—multi-objective question answering (QA)—that extends the number of reasoning steps required to solve a problem. Building on existing multi-turn datasets such as HotpotQA and Natural Questions (Yang et al., 2018; Kwiatkowski et al., 2019), we interleave multiple questions from the original QA corpus and construct a single composite query that requires answering all constituent sub-questions, shown in Prompt 1 of App. B.3. Unlike standard multi-turn QA, this formulation compels the agent to (i) issue multiple search queries, each targeting a distinct sub-question, and (ii) organize the sub-answers into a coherent final response. The augmented dataset inherits the multi-turn retrieval tasks presented in the original HotpotQA. Additionally, to test the agent's long-horizon processing capability, our multi-objective QA combines multiple multi-turn questions into a grand objective and tasks the agent to answer all the questions combined. For instance, the original HotpotQA dataset contains the following two questions: "Which magazine was started first Arthur's Magazine or First for Women?" and "The Oberoi family is part of a hotel company that has a head office in what city?". An augmented task is then: "Answer each of the following questions: "Which magazine was started first Arthur's Magazine or First for Women?" and "The Oberoi family is part of a hotel company that has a head office in what city?" Organize your final answer in <answer> and </answer> and separate the answer to each question with semicolon.

## 4 EXPERIMENTS & RESULTS

We empirically demonstrate the effectiveness of our approach in training the MEM1 agent to perform multi-turn tasks while preserving a near-constant-sized memory state. We evaluate MEM1 against several baselines using a comprehensive set of metrics categorized into *accuracy* (*e.g.,* Exact Match, F1 score, Environment Reward) and *efficiency* (e.g., Peak Token Usage, Dependency Length, Inference Time). All MEM1 variants are fine-tuned from the Qwen2.5-7B Base model (Yang et al., 2024). We use PPO (Schulman et al., 2017) as the RL algorithm as it computes token-level advantages, bringing stability to the training process. While we also experimented with instruction-tuned and

supervised fine-tuned models using curated high-quality trajectories, reinforcement learning from the base model consistently yielded the best performance and generalization.

Our experiments are conducted in two standard environments, each reflecting real-world scenarios that require multi-turn agent interactions. The first environment is question answering with retrieval-augmented generation (RAG) (Kwiatkowski et al., 2019; Yang et al., 2018), where the agent must answer queries by retrieving relevant information from an external knowledge store (either a database or an online search engine). We trained on RAG with a local database (*i.e.,* Wikipedia Corpus) and evaluated on tasks involving open web browsing. For QA, following Sec. 3.3, we construct multi-objective tasks and tested the model performance on tasks with more questions than seen in the training. The second environment is WebShop navigation (Yao et al., 2022), where the agent assists users in online shopping by browsing a website and selecting items based on natural language descriptions. This task requires the agent to iteratively read page content and make navigation decisions, following protocols similar to those in WebGPT (Nakano et al., 2022).

## 4.1 IMPLEMENTATION DETAILS

**Datasets and evaluation metrics.** We train two versions of MEM1 agent for both long-horizon QA and web navigation. For long-horizon QA, we augment the multi-turn QA dataset from (Jin et al., 2025) that mixes data from both HotpotQA (Yang et al., 2018) and Natural Question (Kwiatkowski et al., 2019) to form a multi-objective composite tasks. During training, we use 2-objective task only and test the agent's performance on tasks with more objectives.

For the web agent, we use the WebShop environment (Yao et al., 2022), which also produces a reward during training (Yuan et al., 2025). For all datasets, the train-test split follows the original papers. During RL training, we employ the exact match (EM) metric for QA tasks (details in App. B.4.1) and the environment reward for WebShop (Yao et al., 2022; Yuan et al., 2025). To evaluate the effectiveness of various approaches, we measure the EM and F1 score for QA tasks and final reward for the WebShop environment (Yao et al., 2022; Yuan et al., 2025). To evaluate efficiency, we consider the peak token usage, average dependency, and average inference time. The test datasets are obtained from the original papers which consist of out-of-distribution data. The former two metrics measure the memory efficiency, while the latter measures the time efficiency. The detailed definitions of the metrics are in App. B.4.1. The prompt and format can be found in App. B.3.

**Baselines.** To evaluate the accuracy and efficiency of MEM1, we compare it against two groups of baselines: (i) prior published methods, and (ii) ablations of our approach.

**Prior published baselines.**

- **QA environment:** Search-R1 (Jin et al., 2025), DeepResearcher (Zheng et al., 2025), and the larger-scale model Qwen2.5-14B-Instruct (Yang et al., 2024). Details of Search-R1 and DeepResearcher are provided in App. B.4.2.
- **WebShop environment:** Agent-FLAN (Chen et al., 2024), Agent-R (Yuan et al., 2025), and AgentLM (Zeng et al., 2023).
- **Context compression baseline:** A-MEM (Xu et al., 2025), which augments an Instruct model with a vector database for memory retrieval.

**Ablations of our method.**

- **Truncation baseline (prompt only):** We apply MEM1's agentic truncation prompt template and rollout to an instruct model without RL, isolating the effect of the prompt and rollout design alone. We find that training provides significant performance gains, even though prompt-only rollout already offers some efficiency benefits. Detailed results are presented in Tab. 1, Tab. 2, and Tab. 3.
- **SFT baseline:** Train a supervised fine-tuned model on trajectories curated from GPT-4o (OpenAI, 2024) based on MEM1's rollout, enabling comparison with the RL-trained agent. We find that although supervised fine-tuning improves performance, reinforcement learning is much more effective at enabling generalizability. Detailed results are presented in App. F.1.
- **Memory-Reasoning Coupling**: Explicitly separate memory and coupling in MEM1's internal state to investigate the effect of integrating the two in terms of performance and efficiency. We find that integrated memory and reasoning can benefit both performance and efficiency in our evaluated tasks. Detailed results are presented in App. F.3.

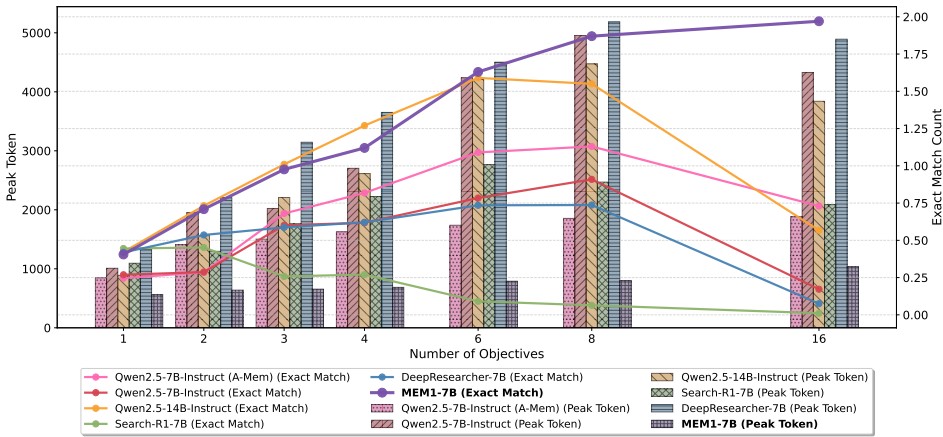

Figure 2: Performance and efficiency scaling of MEM1 (trained on 2-objective QA) with the number of objectives in multi-objective tasks. MEM1 outperforms the other models and baselines while having an almost constant scaling in memory usage. Note that at 16-objective, the context of baseline models does not increase anymore since their model performance has degraded (some collapsed).

Table 1: Comparison of models on multi-objective multi-turn QA tasks. Arrows indicate the desired directions. Numbers in red indicate collapsed model behavior (extremely low performance). (truncate) means using MEM1's prompt and rollout pipeline. (A-Mem) means using MEM1's prompt and rollout pipeline with A-Mem's external memory module (Xu et al., 2025). MEM1-QA means MEM1 trained on 2-objective QA task. Dependency scores are presented in due to limited space.

| Model | 2-Objective | | | | 8-Objective | | | | 16-Objective | | | |
|---|---|---|---|---|---|---|---|---|---|---|---|---|
| | EM ↑ | F1 ↑ | Peak ($\times 10^2$) ↓ | Time (s) ↓ | EM ↑ | F1 ↑ | Peak ($\times 10^2$) ↓ | Time (s) ↓ | EM ↑ | F1 ↑ | Peak ($\times 10^2$) ↓ | Time (s) ↓ |
| Qwen2.5-14B-Inst | **0.732** | **0.902** | 15.6±0.19 | 5.49 ± 0.16 | 1.55 | 1.87 | 44.7 ± 0.37 | 16.2 ± 0.27 | 0.567 | 0.703 | 38.4±0.71 | 29.7±0.75 |
| Qwen2.5-7B-Inst | 0.268 | 0.366 | 19.6±0.33 | 4.60±0.08 | 0.87 | 1.10 | 49.5±0.40 | 13.9±0.18 | 0.165 | 0.213 | 43.3±0.62 | 15.5±0.23 |
| Qwen2.5-7B-Inst (A-MEM) | 0.286 | 0.371 | 14.1±0.10 | 24.6±0.51 | 1.13 | 1.43 | 18.6±0.10 | 53.7±1.26 | 0.730 | 0.961 | 18.8±0.14 | 91.2±2.44 |
| Qwen2.5-7B-Inst (truncate) | 0.262 | 0.336 | 8.28±0.06 | 5.89±0.16 | 0.97 | 1.23 | 11.8±0.10 | 11.9±0.20 | 0.396 | 0.497 | 13.3±0.16 | 22.1±0.60 |
| Search-R1 (original) | 0.452 | 0.531 | 13.0±0.08 | 4.09 ± 0.23 | 0.064 | 0.08 | 24.7 ± 0.19 | **4.25±0.16** | 0.009 | 0.011 | 20.9±0.03 | **4.75±0.18** |
| Search-R1 (trained on 2-obj task) | 0.544 | 0.646 | 13.68±0.23 | 10.60 ± 0.32 | 0.471 | 0.575 | 22.48±0.47 | 19.60±0.50 | 0.520 | 0.647 | 24.8±0.57 | 23.35±0.77 |
| Search-R1 (trained on 2-obj task + truncate) | 0.446 | 0.546 | **6.12±0.04** | 17.5 ± 0.22 | 0.162 | 0.204 | **4.95±0.02** | 21.5±0.18 | 0.091 | 0.107 | **5.28±0.03** | 24.6±0.22 |
| DeepResearcher | 0.536 | 0.650 | 22.0±0.43 | **4.01±0.07** | 0.73 | 0.90 | 51.8±0.35 | 11.3±0.14 | 0.071 | 0.106 | 48.9±0.66 | 15.8±0.19 |
| **MEM1-QA** | 0.709 | 0.838 | 6.40±0.02 | 6.49 ± 0.07 | 1.87 | 2.31 | 8.01±0.06 | 8.68±0.12 | 1.97 | 2.39 | 10.4±0.09 | 8.70±0.12 |

## 4.2 MEM1 ON MULTI-OBJECTIVE MULTI-TURN TASKS

One key advantage of MEM1 agents lies in their efficient management of long-horizon interactions with the environment. To demonstrate this, we train our MEM1 agent with a 2-objective augmentation of the QA dataset, and subsequently test it against other models on held-out multi-objective test sets similarly constructed from the original QA corpus. As elaborated in Sec. 3.3, these multi-objective tasks require substantially more reasoning turns to complete, thus serving as a more demanding benchmark for memory management. As shown in Tab. 1, MEM1 consistently outperforms other 7B counterparts: when tested on 16-objective task, it achieves over $10\times$ higher EM score, while reducing peak context length by more than 70% and cutting inference latency by about one-half. This demonstrates that MEM1 not only improves task success on more complex multi-objective settings, but also does so with markedly better efficiency in both memory and runtime. A visualization of the trend in agent performance as the complexity of tasks increases is shown in Fig. 2. As the number of objectives increases, the Peak Token Usage of all other methods and models scales nearly linearly. In contrast, MEM1 maintains an almost constant peak token count with only a slight increase.

Notably, while MEM1 initially underperforms Qwen2.5-14B-Instruct, its performance gradually catches up as the number of objectives increases, eventually surpassing the 14B model, which has double the parameter count. As shown in Fig. 2, at $1, 2, 3, 4, 6$ objectives, MEM1 has close EM scores compared to the 14B model. As the number of objectives continues to increase, MEM1 achieves significantly higher EM scores. In the 16-objective task, MEM1 achieves over $3\times$ the EM score, while requiring only 27.1% of the peak tokens and 29.3% of the total inference time compared to Qwen2.5-14B-Instruct. This efficiency translates to significantly reduced GPU memory requirements and overall computing resource demands.

Table 2: The experimental results for WebShop. For a fair comparison, we do not report GPT's inference time. For Agent-R, scores are taken from the original paper, as the model is closed source. MEM1-WebShop means MEM1 trained on WebShop environment.

| Model | Avg Final Reward ↑ | Peak Token ($\times 10^3$) ↓ | Dependency ($\times 10^6$) ↓ | Inference Time Per Traj (s) ↓ |
|---|---|---|---|---|
| GPT-4o | 25.48 | $5.30 \pm 1.23$ | $3.99 \pm 1.16$ | *N/A* |
| GPT-4o (truncate) | 13.82 | $0.99 \pm 0.99$ | $0.81 \pm 0.23$ | *N/A* |
| GPT-4o (A-MEM) | 24.50 | $1.84 \pm 0.06$ | $0.31 \pm 0.11$ | *N/A* |
| Qwen2.5-7B-Instruct | 18.42 | $5.64 \pm 1.34$ | $3.38 \pm 0.89$ | $12.31 \pm 1.82$ |
| Qwen2.5-14B-Instruct | 12.34 | $5.44 \pm 0.92$ | $3.30 \pm 0.61$ | $18.17 \pm 2.32$ |
| Agent-FLAN-7B | 40.35 | $3.37 \pm 1.12$ | $2.18 \pm 1.62$ | $9.95 \pm 6.19$ |
| Agent-R-8B | 63.91 | *N/A* | *N/A* | *N/A* |
| AgentLM-7B | 63.60 | $2.24 \pm 0.40$ | $0.28 \pm 0.07$ | $3.91 \pm 1.07$ |
| AgentLM-13B | 70.80 | $2.36 \pm 0.46$ | $0.30 \pm 0.08$ | $5.23 \pm 1.59$ |
| **MEM1-WebShop** | **70.87** | $\mathbf{0.81 \pm 0.10}$ | $\mathbf{0.15 \pm 0.16}$ | $\mathbf{2.61 \pm 0.48}$ |

## 4.3 MEM1 ON SINGLE-OBJECTIVE MULTI-TURN TASKS

While MEM1 is designed to train agents for very long-horizon tasks, our training method also delivers improved capability with existing multi-turn tasks while achieving much greater efficiency at the same time, all without being explicitly trained on the single-objective versions of these tasks. Note that single-objective tasks also require multiple turns of interaction to produce the desired output.

**Long-horizon web navigation in WebShop.** Beyond QA tasks, we further evaluate the effectiveness of MEM1 in managing long-horizon interactions in the form of web navigation. We show the experimental results in Tab. 2. Trained in the WebShop environment (see App. B.6), MEM1 outperforms other agent training baselines, including Agent-Flan, Agent-R, and AgentLM when utilizing models of similar size. Furthermore, MEM1 achieves remarkable efficiency improvements compared to the best baseline method, AgentLM, featuring a $2.8\times$ improvement in Peak Token Usage, a $1.9\times$ improvement in Dependency, and a $1.5\times$ improvement in Inference Time. MEM1 even surpasses AgentLM-13B, a model with twice the parameter count of our trained model. Additionally, our results indicate that using MEM1 is significantly better than OpenAI's GPT-4o on the WebShop tasks, even when the truncation prompt templates or A-MEM techniques are applied to GPT-4o.

**Single-objective QA in Wikipedia.** Tab. 3 presents the accuracy and efficiency metrics for evaluations on single-objective QA tasks on Wikipedia (Jin et al., 2025), where the agent can make retrieval requests from the Wikipedia datastore via RAG. The MEM1 used in this evaluation is the same as the one detailed in Sec. 4.2, which is trained solely on a 2-objective task. Overall, MEM1 demonstrates superior efficiency across all three evaluated efficiency metrics, while simultaneously achieving the highest EM score and an F1 score comparable to that of Qwen2.5-14B-Instruct. This improvement is attributed to the MEM1 agent's ability to consolidate memory from previous interactions into a compact internal state, which reduces the number of tokens used in the context. We also observe that SFT significantly underperforms RL, highlighting the necessity for RL-based training.

**Transfer performance to online Web-QA.** To validate the transferability and generalizability of the trained MEM1 agent, we test MEM1 trained on RAG-QA in an online web-QA environment (Zheng et al., 2025), which is unseen by the agent. This environment is similar to the Wiki-QA task in Sec. 4.3. However, instead of retrieving information from a local Wiki document store, the agent needs to conduct web searches through an API service (e.g., Google Search API) that returns results including titles, snippets, and URLs to answer the QA problems . As shown in Tab. 3, MEM1 consistently exhibited improved efficiency alongside comparable effectiveness in this unseen setting. The result demonstrates that MEM1 has learned actual reasoning and memory management capability rather than overfitting to the RAG dataset.

## 4.4 ANALYSIS ON EMERGENT AGENT BEHAVIORS AND FAILURE CASES

Through analyzing MEM1's multi-turn interaction traces trained on 2-objective QA, we observe a range of emergent behaviors that are critical for handling long-horizon, multi-objective tasks,

Table 3: Transfer performance comparison across environments for out-of-distribution single-objective tasks. Arrows indicate the desired direction. (SFT) means training with SFT and applying MEM1's prompt and rollout. Note that DeepResearcher is specifically trained on the single-objective Online Web-QA task with F1 score as the optimization objective, and Search-R1 is specifically trained on the single-objective Wiki-RAG task with EM as the objective.

| Environment | System | EM ↑ | F1 ↑ | Peak Token ($\times 10^2$) ↓ | Dependency ($\times 10^5$) ↓ | Inference Time ↓ |
|---|---|---|---|---|---|---|
| Wiki RAG | Qwen2.5-7B-Inst (truncate) | 0.287 | 0.382 | $6.28 \pm 0.05$ | $1.65 \pm 0.04$ | $2.26 \pm 0.04$ |
| | Qwen2.5-7B-Inst (A-MEM) | 0.246 | 0.373 | $8.47 \pm 0.12$ | $0.92 \pm 0.03$ | $11.2 \pm 0.40$ |
| | Qwen2.5-7B-Inst | 0.269 | 0.390 | $9.32 \pm 0.19$ | $1.17 \pm 0.04$ | $2.31 \pm 0.04$ |
| | Qwen2.5-14B-Inst | 0.422 | **0.534** | $8.89 \pm 0.21$ | $2.22 \pm 0.10$ | $6.73 \pm 0.24$ |
| | Search-R1 (original) | **0.445** | 0.516 | $11.0 \pm 0.25$ | $1.50 \pm 0.05$ | $\mathbf{2.23 \pm 0.14}$ |
| | DeepResearcher | 0.419 | 0.503 | $13.3 \pm 0.34$ | $7.04 \pm 0.33$ | $3.86 \pm 0.09$ |
| | **MEM1-QA (SFT)** | 0.302 | 0.358 | $6.54 \pm 0.05$ | $3.30 \pm 0.13$ | $4.84 \pm 0.21$ |
| | **MEM1-QA** | 0.405 | 0.471 | $\mathbf{5.63 \pm 0.03}$ | $\mathbf{0.76 \pm 0.02}$ | $3.79 \pm 0.07$ |
| Online Web-QA | Qwen2.5-7B-Inst | 0.334 | 0.451 | $8.37 \pm 0.18$ | $1.39 \pm 0.06$ | $2.20 \pm 0.04$ |
| | DeepResearcher | 0.372 | **0.492** | $10.27 \pm 0.19$ | $2.86 \pm 0.14$ | $2.87 \pm 0.06$ |
| | **MEM1-QA** | **0.397** | 0.485 | $\mathbf{5.79 \pm 0.06}$ | $\mathbf{0.44 \pm 0.02}$ | $\mathbf{1.84 \pm 0.03}$ |

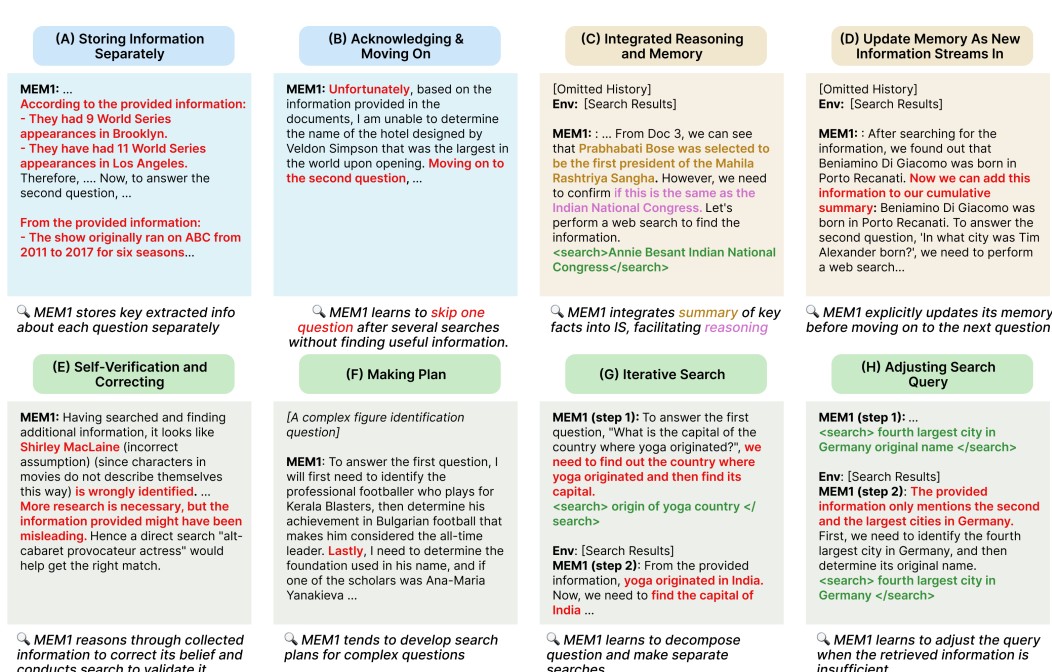

Figure 3: Snippets of internal states and actions showing MEM1's Emergent Behaviors in 2-objective QA tasks. Light Blue denotes behaviors related to multi-objective tasks. Beige denotes behaviors related to memory in internal state. Pastel Green denotes behaviors related to general search strategies.

demonstrating capabilities well beyond simple retrieval. Additionally, we provide detailed qualitative and quantitative analysis of cases where MEM1 fails in App. G First, MEM1 learns to **manage multiple questions concurrently** by maintaining a structured internal state. As shown in Fig. 3(a), when faced with two multi-turn questions, the agent stores and updates memory for each question separately, guiding subsequent searches based on the identified information gaps. In (b), MEM1 exhibits the ability to shift focus when progress on one question stalls, recognizing difficulty and prioritizing the more tractable objective. Meanwhile, MEM1 learns to **interleave reasoning and memory** in its internal state $S$'s, weaving important information into its decision-making process to support both information retention and action selection. In Fig. 3 (c), MEM1 explicitly extracts important information from previous search results and leverages it to formulate the next query that best addresses the current information gap. In addition, (d) shows that when new, relevant information is retrieved, MEM1 explicitly reasons about its significance and selectively updates its memory. We believe that learning these interleaved behaviors is key to achieving efficiency gains

in memory without degrading performance. Beyond behaviors unique to our multi-objective setup and memory architecture, MEM1 also exhibits several **general-purpose search strategies**. In (e), the agent performs self-verification, correcting an earlier misconception and issuing a new query for confirmation. In (f), complex queries are decomposed into manageable subgoals before initiating the search. In (g), for questions requiring multi-turn information gathering, MEM1 extracts key information from search results and uses it to inform the next search. In (h), when overly specific queries fail, MEM1 re-scopes its query to improve retrieval. Notably, many such behaviors, like verification, making a plan, and iterative search, are also reported in recent studies on deep research agents (Jin et al., 2025; Zheng et al., 2025).

## 5    CONCLUSION, LIMITATIONS, AND FUTURE WORK

We introduced MEM1, a reinforcement learning framework that enables language agents to perform long-horizon reasoning with consolidated memory. By integrating inference-time reasoning and memory consolidation into a unified internal state, MEM1 addresses the scalability challenges of prompt growth and achieves competitive performance across QA and web navigation benchmarks, with substantially reduced memory usage and inference latency. Despite these advantages, MEM1 assumes access to environments with well-defined and verifiable rewards. However, many open-ended tasks present ambiguous or noisy reward structures. Fully realizing the potential of MEM1 therefore requires advances in modeling such tasks and designing suitable reward mechanisms—challenges that lie beyond the scope of this paper. A promising future direction is to explore methods for training MEM1 agents in these open-ended settings where reward signals are sparse, delayed, or implicit.

ACKNOWLEDGMENT

This research is supported by the National Research Foundation (NRF), Prime Minister's Office, Singapore under its Campus for Research Excellence and Technological Enterprise (CREATE) programme. The Mens, Manus, and Machina (M3S) is an interdisciplinary research group (IRG) of the Singapore MIT Alliance for Research and Technology (SMART) centre. The computational work for this article was partially performed on resources of the National Supercomputing Centre, Singapore (https://www.nscc.sg).

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

## A    LARGE LANGUAGE MODEL USAGE

We use GPT-5 for both paper writing and literature research. Specifically, we use GPT-5 to help us with proofreading the paper, correcting grammar mistakes, and polishing the writing throughout the paper. We also use GPT-5 to conduct literature research in our related works section.

## B    DETAILS OF MEM1

### B.1    COMPUTING RESOURCES AND TRAINING DETAILS

All trainings of MEM1 are conducted on $4$ H100 or H200 GPUs. We use the veRL framework (Sheng et al., 2024) for RL and Swift (Community, 2024) for SFT. For RL, both the data batch size and mini batch size are set to $64$. Learning rate is set to $10^{-6}$ for the actor model and $10^{-5}$ for the critic model with a linear warmup of $50$ steps. Temperature is set to $1$ during training and $0.01$ during inference.

All evaluations are conducted on a single H200 GPU, which serves the respective models as an API service using the vLLM framework (Kwon et al., 2023) with automatic prefix caching enabled.

### B.2    RAG CONFIGURATION

For RAG on local Wiki corpus, we use Faiss-GPU (Douze et al., 2024) serving an E5 Base model (Wang et al., 2022). The Wiki corpus is taken from a Wikipedia 2018 dump (Karpukhin et al., 2020). The number of passages for each retrieval is set to 3 for a fair comparison with other methods.

For online web search queries, we use Serper API (Serper, 2025), which offers Google search results including titles, snippets, and URLs. For each search, we return the top 10 results to the agent as external information. We do not ask the agent to retrieve the content of specific webpages.

## B.3 PROMPTS

---

**Prompt 1: Multi-Objective Task (QA)**

```
You will answer multiple complex questions using iterative
reasoning, summarization, and web search.

At each step, you will see the questions, a cumulative
summary of relevant information, the current search query,
and search results (except in the first step, where only the
questions are provided). Your task is to:

1. Perform reasoning and update a cumulative, concise
summary within <think> ... </think>. This acts as
persistent memory and must include all essential information
from previous <think> and <information> tags.

2. Then choose one of the following actions:
- If any question remains unanswered, issue a single query
for one question inside <search> ... </search>. The query
should consist of keywords or a short phrase. Only search
one question at a time.
- If all questions are answered, provide the final
answers--separated by semicolons--within <answer> answer1;
answer2; ... </answer>. The answers must be concise,
contain only essential words, and avoid any explanations.

Important:
- Always follow this structure after <information> or
the initial questions: <think> ... </think><search> ...
</search> or <think> ... </think><answer> ... </answer>.
- Do not search multiple queries or questions simultaneously.

Answer the following questions:[QUESTIONS]
```

**Prompt 2: Single-Objective Task (QA)**

```
You will answer a complex question through iterative
reasoning, summarization, and web searches.

At each step, you can see the question, previous summary
in <think> ...  </think>, search query in <search> ...
</search>, and the returned information in <information> ...
</information> (except the first step where you will be given
only the question).  Then, you should:

1.  Conduct reasoning, and then update a concise, cumulative
summary with essential information inside <think> </think>.
This is your persistent memory and should include all
important information from previous <think> </think> and
<information> </information> (i.e.  information and answers
already found for questions).

2.  Then choose one:
- Issue a query (i.e., key words / phrases for search)
inside <search> </search> (you may search repeatedly until
the answer is clear).  This query will be used to conduct
search and return the results in <information> results
</information>
- Provide the final concise answer (no explanations) if no
additional information is needed inside <answer> </answer>.
The answer should be concise and only contain the words
necessary to answer the question.

After <information> </information> (or question at the
beginning), you should always follow the order:  <think>
...  </think><search> ...  </search> or <think> ...
</think><answer> ...  </answer>.

Question:  [QUESTION]
```

---

**Prompt 3: Single-Objective Task (WebShop)**

```
You are browsing an online shop.  Your goal is to find
a product that matches the given description.  You will
interact with the site step-by-step.  Each step gives you
a <state>...</state> representing the current webpage.  You
must decide what action to take next until you identify the
correct product.

Available actions (shown in the <state> tag) depend on the
page:
- On the search page:  search[<keywords>]
- On search result pages:  click[<item url>] to view a
product, or click[next >] to go to the next results page
- On product pages:  click[description], click[features],
click[color], click[size], click[buy now]
- To return to search:  click[back to search]

Example goal:  "Find a gingko light and 20x20 pillow
cover that is hand painted." Example first action:
<answer>search[gingko light 20x20 pillow cover hand
painted]</answer> Only respond with valid actions formatted
as:  search[...], click[...], etc.

After you navigate and find the product that best fits the
user goal, you should click[buy now] to buy the product at
the product page when the buy now button is available.

Product Description:  [PRODUCT DESCRIPTION]
```

---

## B.4 Implementation Details of Metrics and Baselines

### B.4.1 Metrics

**Exact match.** In QA tasks, we use exact match (EM) as both the verifiable reward for the RL pipeline and the evaluation metric for the final output. The final response is extracted from between <answer> and </answer>. In multi-objective settings, the response should contain answers to each question separated by semicolons. If the XML tags are mismatched, or if the number of provided answers does not correspond to the number of questions, a score of $0$ is assigned. Otherwise, $1$ point is credited for each correct answer. During RL training, we do not provide any other intermediate rewards or format penalties, as we find that such manual interventions can interfere with the agent's learning process (see more in Sec. 4.4).

**F1 score.** The F1 score computes the harmonic mean between the precision $p$ and recall $r$. In the case of string matching, we split both the predicted answer and the ground truth. For example, if the ground truth is "United States of America", it is split into a list with lower-case words: "united", "states", "of", "america". The same works for the predicted answer. Then, denote the number of common words as $c$. Further denote the number of words in the predicted answer as $l$ and the number of words in the ground truth as $g$. Then, precision is calculated as $p := c/l$ and recall is calculated as $r := c/g$. The F1 score is finally computed as

$$\text{F1} := 2 \times \frac{p \times r}{p + r} \, .$$

If multiple ground truths are present, the maximum of all F1 scores is chosen. For multi-objective tasks, the final F1 is the sum of the F1 scores for each sub-question.

**Peak token usage.** Peak token usage is calculated as the maximum number of tokens (using GPT-4o-mini tokenizer) in any single sequence throughout the agent's entire trajectory. For fair

comparison in our experiments, the system prompt is excluded when computing this sequence length. The peak token usage serves as a proxy for the inference-time memory requirement.

**Dependency length.** Following (Zhang et al., 2025), the dependency metric is defined as the total number of historical tokens on which each generated token effectively depends. Let $T$ denote the total number of interaction steps. For each step $i \in [T]$, let $n_p^{(i)}$ be the number of prefix tokens and $n_o^{(i)}$ be the number of output tokens generated. The dependency metric is then calculated as

$$\texttt{Dependency} := \sum_{i \in [T]} \frac{(2n_o^{(i)} + n_p^{(i)}) \times n_o^{(i)}}{2} .$$

At a high level, this metric quantifies the cumulative computational cost associated with the generation of an output trajectory. It is important to note that in MEM1, prefix tokens from previous steps are consolidated into a new internal state, rather than being continuously accumulated. In our experiments, we ignore the tokens in the system prompt when calculating the dependency metric.

**Inference time.** Inference time for each trajectory is recorded as the total elapsed time required to generate the complete output trajectory. For all experiments, these measurements are conducted on a single H200 GPU, operating with 10 concurrent threads. The vLLM inference framework is utilized, with its automatic prefix caching feature enabled.

### B.4.2 BASELINES

**Search-R1.** As detailed in (Jin et al., 2025), the model is trained on the 1-objective task with the same dataset as MEM1. Search-R1 also uses exact match as its reward function. In comparison, MEM1 is trained exclusively on 2-objective tasks.

**Deep Researcher.** As detailed in (Zheng et al., 2025), the model is trained on 1-objective task with a curated set from various QA datasets including HotPotQA and Natural Questions. Deep Researcher adopts the F1 score as the reward function.

### B.5 ALGORITHM

We provide an outline of the rollout of MEM1, which actively manages its context in Alg. 1. Parts of the pseudo-code follow (Jin et al., 2025). We follow (Wei et al., 2022; DeepSeek-AI et al., 2025) and annotate each component using XML-style tags: `<IS>` for internal state (reasoning $S_t$), `<query>` for environment queries $A_t, t < T$, `<answer>` for the agent's responses $A_T$, and `<info>` for external observations or tool outputs $O_t$.

---

**Algorithm 1** MEM1 Rollout

---

**Require:** Task prompt $x$, policy model $\pi_\theta$, world model $\mathcal{W}$, maximum turn $T$
**Ensure:** Final response $y$
1: Initialize rollout sequence $y \leftarrow \varnothing$
2: Initialize turn count $t \leftarrow 0$
3: **while** $t < T$ **do**
4:     Initialize current policy rollout sequence $y_t \leftarrow \varnothing$
5:     **while** True **do**
6:         Generate response token $y_r \sim \pi_\theta(\cdot \mid x, y + y_t)$
7:         Append $y_r$ to rollout sequence $y_t \leftarrow y_t + y_r$
8:         **if** $(t = T - 1)$ and $y_r \in [\texttt{</answer>}, \texttt{<eos>}]$ **then**
9:           **break**                          // prevent the agent from searching further
10:         **else if** $y_r \in [\texttt{</query>}, \texttt{</answer>}, \texttt{<eos>}]$ **then**
11:           **break**
12:         **end if**
13:     **end while**
14:     $y \leftarrow y_t$                                       // all previous context removed.
15:     **if** $\texttt{<query>}$ $\texttt{</query>}$ detected in $y_t$ **then**
16:         Extract search query $q \leftarrow \text{Parse}(y_t, \texttt{<query>}, \texttt{</query>})$
17:         Retrieve environment feedback $d \leftarrow \mathcal{W}(q)$ from local storage, Search engine, HTML, $\cdots$
18:         $\texttt{HINT} \leftarrow \texttt{You have } \{T - t\} \texttt{ turns left.}$
19:         Insert $d$ into rollout $y \leftarrow y + \texttt{<info>}\texttt{HINT} + d\texttt{</info>}$
20:     **else if** $\texttt{<answer>}$ $\texttt{</answer>}$ detected in $y_t$ **then**
21:         **return** final generated response $y$
22:     **else**
23:         Mark the sample as invalid
24:     **end if**
25:     Increment turn count $t \leftarrow t + 1$
26: **end while**
27: **return** final generated response $y$

---

### B.6 MEM1 on Webshop Training Details

We use the same rollout pipeline and policy update mechanism for training MEM1 on WebShop. Compared to the QA tasks, we use a tailored prompt that retains the gist of memory consolidation with instructions specific to the WebShop environment, as shown in Prompt 3. Another distinction is that the WebShop environment comes with its own reward function corresponding to each state. Therefore, we do not use exact match but the built-in reward function as the reward signal when training in WebShop environment. The training and test splits also follow the original paper (Yao et al., 2022), with the first 1000 samples as the test set, the 1000th to 1500th as the val set, and the remaining as the train set.

### B.7 Additional Discussion on the Attention Matrix Design.

We wish to note that our modification to the attention matrix *does not* fully recover the attention of the original trajectories because of the change in position ids. Specifically, prior works (Leviathan et al., 2023; Chen et al., 2023; Cai et al., 2024) that utilized the attention matrix to compress multiple trajectories mainly targeted tree-exploration, i.e., generating multiple sequences with the same prefix. For these works, on top of the attention matrix, they adjusted the position ids as well, so each trajectory follows a consecutive increasing position ids. However, in MEM1, the prefix does not remain the same because of memory consolidation. This results in each $\texttt{<IS>}$ having two possible position ids, one for the previous turn and one for the next turn. To completely recover the original attention, we need to duplicate each $\texttt{<IS>}$ and assign different position ids to the two copies. However, such duplication can significantly slow down training because the training trajectories are now much longer.

As such, for training efficiency, we do not duplicate the $\texttt{<IS>}$ and assign the position ids for the previous trajectory to each $\texttt{<IS>}$. While this modification slightly deviates from the "ideal"

implementation, effectively, it can be viewed as simply adding white spaces in the training trajectories and has no significant impact on the experimental results.

## C    BROADER IMPACTS

MEM1 opens up the potential to enable more scalable, efficient, and intelligent AI agents capable of sustaining long, goal-directed interactions in dynamic environments. As AI systems are increasingly deployed in complex real-world tasks—such as scientific research, legal analysis, personalized education, and digital customer service—models must go beyond single-turn capabilities and manage evolving contexts over many steps. MEM1's memory-consolidation mechanism allows language models to maintain high performance without the growing computational and environmental costs typically associated with long-context processing. By reducing inference-time memory and compute demands, MEM1 paves the way for more sustainable and scalable AI deployment, making advanced reasoning agents accessible to a wider range of users and institutions, including those with limited resources. Moreover, MEM1's unified framework of reasoning and context consolidation sets a precedent for future research on intelligence that can learn to adapt, reflect, and summarize information autonomously, inspiring more trustworthy, interpretable, and human-aligned AI systems.

## D    TRAINING TRAJECTORY ANALYSIS OF MEM1

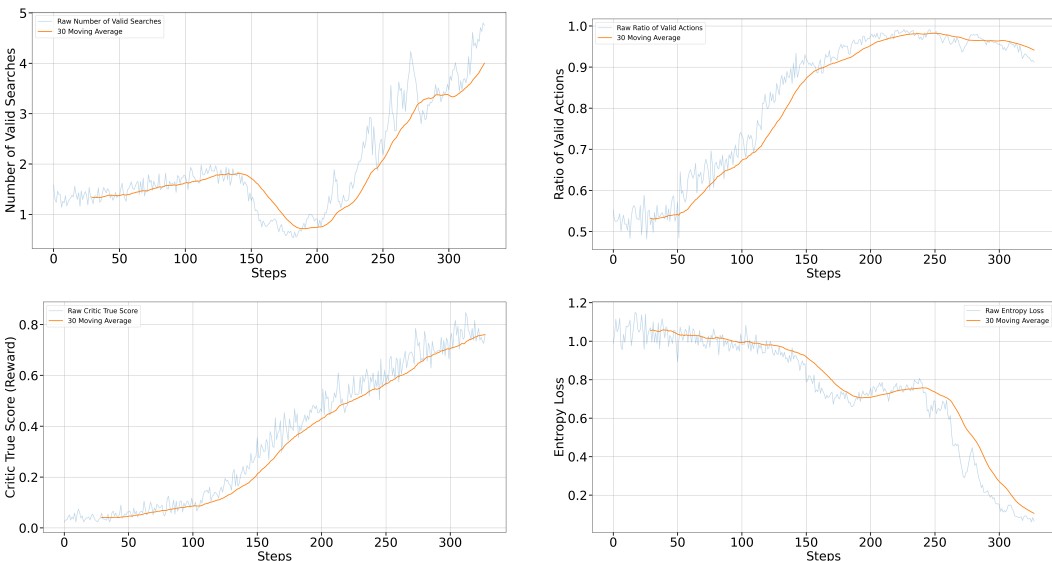

Figure 4: Metrics of training progresses for MEM1 with RL.

We present the training dynamics of the 2-objective QA-trained MEM1 in Fig. 4, where several distinct phases emerge during the learning process. In the initial exploration phase (first 50 steps), the agent demonstrates little task proficiency. The reward remains consistently low, while the entropy loss is high, suggesting random or undirected behavior. The ratio of valid actions hovers around 0.55, indicating that the agent frequently fails to follow the expected output format. During this period, MEM1 has not yet learned to reliably use the required structure involving `<query>` and `<answer>` tags.

Shortly after, we observe the onset of format acquisition. The agent gradually improves its structural consistency, reflected in the rising ratio of valid actions. This improved adherence to format correlates with an increase in reward, suggesting that proper formatting directly contributes to the agent's task success. By around step 150, a notable behavioral shift occurs. The number of valid searches begins to drop sharply, while the reward continues to increase. This implies that the agent has discovered a shortcut: by reducing the number of searches—perhaps to avoid format violations—it can maintain high format fidelity and improve its reward without fully solving the task. This short-horizon

optimization suggests the agent is exploiting the reward structure, favoring formatting compliance over content completeness.

Between steps 150 and 200, the agent enters a phase of refined format mastery. The ratio of valid actions steadily climbs, but the number of searches remains low. During this phase, reward growth slows, and entropy begins to flatten. The plateau in entropy indicates that the agent is looking for new policies to boost the reward. At this stage, the agent has reached a local optimum: it's producing valid but under-informed answers.

After step 200, a second behavioral shift occurs. The number of valid searches begins to rise again, suggesting that the agent is learning to extend its interaction horizon to gather more information. The agent learns to balance formatting constraints with information acquisition. As a result, the reward increases more sharply. Finally, after step 250, the agent enters a phase of policy consolidation. The entropy loss drops sharply—signaling a transition from exploration to exploitation—as the agent settles into a more deterministic, high-reward policy. By this stage, the agent effectively combines format compliance, sufficient searching, and high-quality answer generation.

# E  ADDITIONAL RESULTS

## E.1  RESULTS ON THE DEPENDENCY METRIC

Due to the limited page width, we present the full dependency metric, as defined in App. B.4.1, results for Tab. 1 in Tab. 4. From these results, we observe that MEM1 consistently achieves the lowest dependency across all task settings. Although Search-R1 exhibits dependency close to those of MEM1, its performance degrades substantially on more complex tasks, making its efficiency metric less informative.

Table 4: Additional dependency score results for MEM1 ($\downarrow$ lower is better). Dependency values (mean $\pm$ standard deviation) are reported in units of $\times 10^6$. The lowest and second lowest values in each column are shown in **bold**.

| Model | 2-Objective | 8-Objective | 16-Objective |
|---|---|---|---|
| Qwen2.5-7B-Instruct (Amem) | $0.401 \pm 0.280$ | $1.440 \pm 0.959$ | $1.697 \pm 1.819$ |
| Qwen2.5-7B-Instruct (truncate) | $0.573 \pm 0.540$ | $1.535 \pm 1.134$ | $2.060 \pm 2.509$ |
| Qwen2.5-7B-Instruct | $0.877 \pm 1.028$ | $4.135 \pm 2.609$ | $4.411 \pm 3.020$ |
| Qwen2.5-14B-Instruct | $0.297 \pm 0.269$ | $1.462 \pm 1.021$ | $4.451 \pm 3.676$ |
| Search-R1 | $\mathbf{0.173 \pm 0.154}$ | $\mathbf{0.601 \pm 0.490}$ | $\mathbf{0.784 \pm 0.902}$ |
| Search-R1 (2-objective trained) | $1.180 \pm 0.919$ | $5.800 \pm 2.316$ | $6.450 \pm 2.909$ |
| Search-R1 (2-objective + truncate) | $1.240 \pm 0.918$ | $5.950 \pm 2.318$ | $6.700 \pm 2.910$ |
| DeepResearcher | $0.799 \pm 0.919$ | $3.891 \pm 2.316$ | $4.355 \pm 2.909$ |
| **MEM1-QA** | $\mathbf{0.177 \pm 0.104}$ | $\mathbf{0.486 \pm 0.332}$ | $\mathbf{0.724 \pm 0.405}$ |

## E.2  ANALYSIS OF THE LENGTH OF THE INTERNAL STATE WITH RESPECT TO THE NUMBER OF INTERACTION TURNS

To examine how the agent's internal state evolves through reasoning, we conduct a subsample analysis on MEM1-QA using 2-objective tasks with up to 6 turns and 8-objective tasks with up to 15 turns. As shown in Fig. 5, although MEM1 does not explicitly regulate internal-state length, we observe no substantial growth across turns: the agent consistently discards unhelpful information and retains only what is necessary. Because the final answers in these QA tasks are short, the agent does not require a continually expanding internal state to converge on correct solutions. While 8-objective tasks naturally produce somewhat longer internal states, the overall size remains stable across interaction turns.

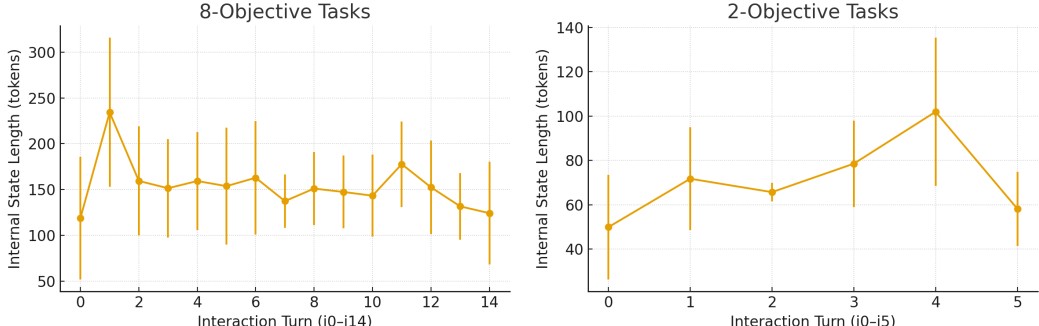

Figure 5: Average internal state length (with standard deviation) versus interaction turn for MEM1-QA across 2-objective (right) and 8-objective (left) tasks, shown using the combined figure.

# F    ANALYSIS ON IMPLEMENTATION DETAILS

## F.1    RL GENERALIZES BETTER THAN SFT

A natural question arises: can Supervised Fine-Tuning (SFT) with high-quality trajectories match the performance of reinforcement learning (RL)? To investigate this, we compare MEM1-QA trained via RL against MEM1-QA (SFT), where both models are trained on the 2-objective QA task. Additionally, the SFT model is further trained on 1-objective and 3-objective QA tasks to enhance its generalization ability. As shown in Tab. 5, the SFT model consistently underperforms compared to its RL counterpart across tasks with varying numbers of questions (objectives). Notably, when the number of objectives exceeds six, the performance of the SFT model collapses, whereas the RL-trained model continues to demonstrate strong robustness and scalability.

Table 5: Comparison of RL and SFT on increasing number of multi-turn questions. Exact match scores ↑ is better. Gap shows absolute difference. Red numbers show collapsed SFT behavior.

| #Q | RL ↑ | SFT ↑ | Gap ↑ | RL Gain (%) ↑ |
|---|---|---|---|---|
| 1 | 0.410 | 0.300 | 0.110 | +36.7% |
| 2 | 0.709 | 0.433 | 0.276 | +63.7% |
| 3 | 0.976 | 0.648 | 0.328 | +50.6% |
| 4 | 1.120 | 0.626 | 0.494 | +78.9% |
| 6 | 1.630 | 0.088 | 1.542 | +1752% |
| 8 | 1.870 | 0.027 | 1.843 | +6826% |
| 16 | 1.900 | 0.000 | 1.900 | — |

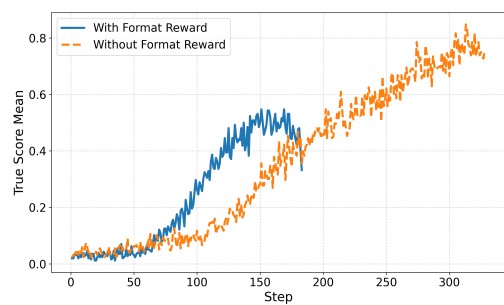

Figure 6: Training curves comparing MEM1 trained with and without format reward.

## F.2    FORMAT REWARD ACCELERATES CONVERGENCE BUT DEGRADES FINAL PERFORMANCE

It is common to incorporate format reward when training reasoning models and multi-turn reasoning agents DeepSeek-AI et al. (2025); Zheng et al. (2025); Jin et al. (2025). In our study, we experimented with a format reward that enforces the agent to produce outputs using specific structural tags: `<IS>`, `<query>`, and `<answer>`. If the agent fails to use the expected tags correctly, the turn is terminated and a penalty of -1 is applied.

As shown in Fig. 6, using the format reward leads to faster convergence during training but results in worse final performance. The format-constrained agent achieves an exact match score of 0.466, compared to 0.709 for MEM1 trained with only outcome-based reward on the same testing set for the 2-objective QA task. Additionally, the format-constrained agent generates fewer tokens, with an average peak of 514.9 tokens, whereas the outcome-reward-trained MEM1 reaches an average peak of 640 tokens.

We hypothesize that the format reward accelerates structural learning but constrains exploration of effective reasoning strategies. As a result, the agent learns to produce shorter responses with valid syntax but develops less effective internal state representations, leading to degraded task performance.

### F.3    COUPLING MEMORY AND REASONING BENEFITS BOTH PERFORMANCE AND EFFICIENCY

We explored whether separating memory and reasoning could simplify the agent design. In this ablation setup, we replaced the integrated internal state with explicit `<summary></summary>` and `<reasoning></reasoning>` tags and trained the model using the same rollout procedure and reinforcement learning algorithm. As shown in Tab. 6, coupling memory and reasoning in a unified internal state consistently yields higher Exact Match scores and substantially lower peak token usage across 2-objective, 8-objective, and 16-objective tasks. These results suggest that maintaining an integrated internal state enables more efficient reasoning and better information retention, ultimately benefiting both performance and computation.

Table 6: Ablation study comparing integrated versus separated memory and reasoning. Peak token counts are reported in units of $\times 10^2$, with standard deviations in parentheses.

| MEM1-QA (integrated internal state) | 2-Objective | 8-Objective | 16-Objective |
|---|---|---|---|
| Exact Match (EM) | 0.709 | 1.87 | 1.97 |
| Peak Tokens ($\times 10^2$) | 6.4 (0.02) | 8.01 (0.06) | 10.4 (0.09) |
| **MEM1-QA (separate memory + reasoning)** | **2-Objective** | **8-Objective** | **16-Objective** |
| Exact Match (EM) | 0.686 | 1.78 | 1.92 |
| Peak Tokens ($\times 10^2$) | 11.8 (0.05) | 9.63 (0.07) | 13.37 (0.11) |

## G    FAILURE CASE ANALYSIS

We analyze the failed trajectories from the two-question setting to understand the limitations of the model and identify recurrent patterns that hinder reliable multi-step reasoning. After careful examination, we identify five representative ways in which the model fails to answer all questions correctly. Overall, three of these five failure modes are related to general limitations of the search agent, while the remaining two arise directly from our iterative compression rollout and compositional tasks. We name them entity ambiguity, hallucination instead of search, logical fallacies, insufficient query refinement, and overwriting correct answers due to loss of the original reasoning path. In Tab. 7, we provide full explanations for these failure types, an illustrative example containing original evidence from the trajectory, and the corresponding frequency.

There are also some non-typical failures that do not fall under these five categories, and we group them as "others." Some failures are difficult to avoid because the Wikipedia corpus we use as the information source may not contain answers for all questions, and some failures stem from formatting issues. We further compute the percentage of each failure type by labeling the error category of every trajectory. Among the 1000 trajectories we examined, where the model answered one question correctly for 397 tasks and both questions correctly for 156 tasks, the percentages and counts for each failure mode are also reported in Tab. 7. Note that a single failed trajectory may contain multiple failure types, so the total count does not necessarily match the number of failed trajectories exactly, but it provides a useful reference for understanding the distribution of failure modes.

| Failure Mode | Explanation | Example | Statistics |
|---|---|---|---|
| **Entity Ambiguity** | The agent chooses the wrong real-world entity among several with the same or similar name and builds its answer on that mistaken identity. | The query asks for the birthplace of the performer of *"Step Into The Light (Myra Song)"*. Search results contain two different "Myra" entities: (1) the Mexican-American pop singer who performs the song, and (2) Myra Taylor, an unrelated jazz singer whose biography lists a birthplace. The model selects Myra Taylor and outputs *"Bonner Springs, Kansas"*, which is irrelevant. | **164 (20.1%)** |
| **Hallucination Instead of Search** | The agent invents facts instead of searching or acknowledging uncertainty. | For *Joni Jenkins* and *Martin Shapiro*, the agent fabricates occupations ("interior designer", "lawyer") and claims it already "recalled" them. The true occupations are politician and screenwriter. | **391 (47.9%)** |
| **Logical Fallacies** | The agent uses invalid or irrelevant criteria to compare entities and draw conclusions. | Asked which director is older, it compares film release years (2005 vs 2008) rather than birth dates, assumes similar directing ages, and concludes the earlier film's director is older. A snippet: "Assuming both directors were of similar ages... the director of *Flying Boys* is older since it was released earlier." | **31 (3.8%)** |
| **Insufficient Query Refinement** | The agent repeats low-yield queries without adapting after irrelevant results. | For "In what country is Awe?", it repeats the exact same query three times and keeps retrieving pages about the emotion "awe" or River/Bridge Awe in Scotland. For "Mrákotín", it repeatedly gets unrelated results (e.g., Mrkonjić Grad) without refining the search strategy. | **35 (4.3%)** |
| **Overwriting Correct Answers Due to Loss of Reasoning Path** | The agent remembers the answer but not the reasoning steps that justified it. | For the question pairing King Midas and Prince's 1984 song, the agent first finds *Purple Rain* but later, lacking the full reasoning chain, focuses on genre metadata and incorrectly claims "no specific type of rain" is mentioned, replacing the correct answer with "power ballad." | **7 (0.9%)** |
| **Others** | Miscellaneous failures not captured by the five modes. | — | **188 (23.4%)** |

Table 7: Summary of failure modes, explanations, examples, and statistics from the two-question setting.

