# OpenReview forum: "MEM1: Learning to Synergize Memory and Reasoning for Efficient Long-Horizon Agents"
_ICLR.cc/2026/Conference — ICLR 2026 Poster_

### Official Review · Reviewer_CDp4 · 2025-10-26

**Soundness:** 3
**Presentation:** 3
**Contribution:** 3
**Rating:** 6
**Confidence:** 5

**Summary:**

The authors introduce MEM1, an end-to-end reinforcement learning (RL) framework that enables agents to update a compact, shared internal state at each turn. This state jointly supports memory consolidation and reasoning by integrating essential prior information with new observations while strategically discarding irrelevant data. The framework is trained using RL with a 2D attention mask to manage the dynamic context during policy optimization. Experiments were conducted in three domains: internal retrieval QA, open-domain web QA, and WebShop navigation. The authors introduced a "multi-objective QA" task, augmenting datasets like HotpotQA by composing multiple questions. Results shows that Mem1 achieves competitive performance across QA and web navigation benchmarks with substantially reduced memory usage and inference latency.

**Strengths:**

1. The paper is clearly written, well-structured, and strongly motivated.
2. On extra-long horizon tasks (16 objectives), MEM1 demonstrates both efficiency and strong performance, indicating that it effectively mitigates the “forgetting” problem in long-context scenarios. Its ability to generalize from training on 2-objective tasks to 16-objective tasks is particularly impressive, suggesting that the model has acquired a universal memory management capability rather than task-specific heuristics.
3. A key novelty of this paper lies in integrating memory management directly into the model’s reasoning process, instead of relying on external memory modules. This design allows the agent to learn what to remember as part of its inheent decision-making strategy, which represents an elegant and conceptually unified approach to handling memory within large reasoning models.

**Weaknesses:**

1. The paper lacks evaluation on several representative multi-hop reasoning benchmarks, such as BrowseComp, GAIA, and WebWalker QA. Including these datasets would provide a more comprehensive assessment of MEM1’s capabilities, particularly in extra-long-horizon tasks that require complex, multi-step reasoning.

2. Incomplete analysis of the Internal State.

(1)	The paper does not provide sufficient analysis of the Internal State mechanism. It would be helpful for the authors to report statistics such as the average length of the Internal State during inference, and to clarify whether its size can be explicitly controlled, or whether it grows dynamically with the number of interaction turns.

(2)	Furthermore, the authors should discuss how the model mitigates hallucinations or inconsistencies in the Internal State representation. Presenting a few failure cases or qualitative examples where the Internal State leads to incorrect reasoning would provide valuable insight into the model’s limitations and potential areas for improvement.

3. The authors claim O(1) context efficiency, but this only accounts for the attention cost. Internal State at step t must include all historical information, so its length and generation cost increase with the task complexity at each step. The authors should report the total FLOPS of MEM1 and compare with other methods (DeepResearcher, SearchR1).

**Questions:**

1. The tasks used in this paper are based on a final outcome reward setting. It would be interesting to explore how the model’s performance changes when applying RLVR algorithm, such as GRPO.
2. If a model incorrectly summarizes or omits a key fact, that fact is permanently lost. How to solve this problem?
3. It could guide the model to use a summarization tool to condense all historical information into a fixed context window. What advantages does MEM1 offer compared to this method?
4. Do different search providers (such as Google Seach and Bing Search) have an impact on performance? What is the search parameter?

---

> ### Author Response · Authors · 2025-11-26
> **Response Part 1**
>
> We thank the reviewer CDp4 for taking time to provide the review and for comenting that our paper is clearly written, well structured, effective, and novel. We address your concerns and questions in the following.
>
> > The paper lacks evaluation on several representative multi-hop reasoning benchmarks, such as BrowseComp, GAIA, and WebWalker QA. Including these datasets would provide a more comprehensive assessment of MEM1’s capabilities, particularly in extra-long-horizon tasks that require complex, multi-step reasoning.
>
> We thank the reviewer for suggesting the more difficult benchmarks. We note that solving these tasks usually demands high reasoning capability of the base model which can be hardly learned purely from RL. As such, we used the concatenation task in our paper to highlight the strength of our proposed method.
>
> To facilitate a deeper understanding of MEM1, we are currently running an experiment to test whether MEM1 can be applied to DeepDive [1], a dataset that incorporates several dependent sub-problems, similar to the benchmarks like browsecomp and webwalker QA. We follow a similar strategy as presented in the original paper to train a base 7B model with SFT. Different from them, we use the trajectories from our MEM1 rollout for SFT. Due to the considerable amount of engineering to adapt our framework to this pipeline as well as the need to collect trajectories from large models, we are still training the model. We will update the results by the end of the rebuttal session. We appreciate the understanding of the reviewer for the delay.
>
> [1] Lu et al., DeepDive: Advancing Deep Search Agents with Knowledge Graphs and Multi-Turn RL, arXiv 2509.10446.
>
> > The paper does not provide sufficient analysis of the Internal State mechanism. It would be helpful for the authors to report statistics such as the average length of the Internal State during inference, and to clarify whether its size can be explicitly controlled, or whether it grows dynamically with the number of interaction turns.
>
>
> We thank reviewers for this helpful suggestion. Our MEM1 method does not explicitly control the length of the internal state. Prompting or a different reward design should help control the length of the internal state more explicitly. However, in our experiments, we do not see a drastic growth in the length of the internal state because useless information is discarded at each turn and only useful information is retained. To better explain this, the eventual answers are not long, so for QA tasks we do not need a progressively growing internal state in order to find the answers. We analyzed the average length of the internal state for each interaction turn for 2-objective tasks with 6 turns and 8-objective tasks with 15 turns, and although 8-objective tasks tend to have longer internal states, we can see that the internal state’s size does not grow that much with the number of interaction turns.
>
> ### Internal State Length vs. Number of Interaction Turns
>
> #### 8-Objective
>
> | Turn | Avg Length | Std Dev |
> |------|------------|---------|
> | i0   | 118.93     | 66.99   |
> | i1   | 234.33     | 81.49   |
> | i2   | 159.39     | 59.46   |
> | i3   | 151.43     | 53.76   |
> | i4   | 159.31     | 53.31   |
> | i5   | 153.73     | 63.88   |
> | i6   | 162.88     | 61.82   |
> | i7   | 137.36     | 29.19   |
> | i8   | 151.08     | 39.84   |
> | i9   | 147.38     | 39.74   |
> | i10  | 143.28     | 44.71   |
> | i11  | 177.75     | 46.75   |
> | i12  | 152.50     | 51.26   |
> | i13  | 131.67     | 36.27   |
> | i14  | 124.21     | 56.11   |
>
> ---
>
> #### 2-Objective
>
> | Turn | Avg Length | Std Dev |
> |------|------------|---------|
> | i0   | 49.89      | 23.56   |
> | i1   | 71.72      | 23.17   |
> | i2   | 65.67      | 4.19    |
> | i3   | 78.50      | 19.50   |
> | i4   | 101.91     | 33.43   |
> | i5   | 58.17      | 16.75   |

---

> ### Author Response · Authors · 2025-11-26
> **Response Part 2**
>
> > Furthermore, the authors should discuss how the model mitigates hallucinations or inconsistencies in the Internal State representation. Presenting a few failure cases or qualitative examples where the Internal State leads to incorrect reasoning would provide valuable insight into the model’s limitations and potential areas for improvement.
>
> We thank reviewers for asking about the failure cases, and we believe a comprehensive analysis of failure cases can give us more insights into the limitations of the proposed method and offer insights for future improvements. For a cleaner presentation, we refer to our response to the reviewer zPnP for details of the failure case analysis.

---

> ### Author Response · Authors · 2025-11-26
> **Response Part 3**
>
> > The authors claim O(1) context efficiency, but this only accounts for the attention cost. Internal State at step t must include all historical information, so its length and generation cost increase with the task complexity at each step. The authors should report the total FLOPS of MEM1 and compare with other methods (DeepResearcher, SearchR1).
>
> We thank the reviewers for paying attention to the complexity discussion and for the helpful suggestion. The **dependency** metric, defined in [1] as the total number of historical tokens on which each generated token effectively depends, measures total computation cost **in a way similar to FLOPs** for models with comparable parameter sizes. We apologize for not including the dependency scores in the main table; we now present them in the updated table below. From the table, we can see that MEM1 consistently achieves the lowest dependency score compared to other baselines. While Search-R1 has a dependency score close to that of MEM1, its performance drops drastically on more complex tasks, making its efficiency score less informative. We will also update the table in the main text accordingly.
>
> | Model                             | 2-Objective Dependency (×10⁶) | 8-Objective Dependency (×10⁶) | 16-Objective Dependency (×10⁶) |
> |----------------------------------|-------------------------------|--------------------------------|---------------------------------|
> | Qwen2.5-7B-Instruct (Amem)       | 0.401344                      | 1.440281                       | 1.697043                        |
> | Qwen2.5-7B-Instruct (truncate)   | 0.572892                      | 1.535069                       | 2.059680                        |
> | Qwen2.5-7B-Instruct              | 0.876792                      | 4.134750                       | 4.410514                        |
> | Qwen2.5-14B-Instruct             | 0.297367                      | 1.461738                       | 4.451320                        |
> | Search-R1                        | 0.173364                      | 0.601225                       | 0.783620                        |
> | DeepResearcher                   | 0.798718                      | 3.891144                       | 4.354958                        |
> | MEM1-QA           | 0.176652                      | 0.485936                       | 0.724110                        |
>
>
> [1] Zhang, J., Zhu, Y., Sun, M., Luo, Y., Qiao, S., Du, L., ... & Zhang, N. (2025). Lightthinker: Thinking step-by-step compression. arXiv preprint arXiv:2502.15589.
>
> > The tasks used in this paper are based on a final outcome reward setting. It would be interesting to explore how the model’s performance changes when applying RLVR algorithm, such as GRPO.
>
> We thank the reviewer for the suggestion. We wish to clarify that our PPO training is also an RLVR algorithm. We opt for PPO training as it is more stable under limited compute than GRPO, especially since we do not have sufficient compute to support a large group size (n > 32). This has also been demonstrated in prior work on reinforcement learning for search agents [1], where GRPO and PPO were compared and PPO was shown to perform better overall.
>
> [1] Jin, B., Zeng, H., Yue, Z., Yoon, J., Arik, S., Wang, D., ... & Han, J. (2025). Search-r1: Training llms to reason and leverage search engines with reinforcement learning. arXiv preprint arXiv:2503.09516.
>
>
> > If a model incorrectly summarizes or omits a key fact, that fact is permanently lost. How to solve this problem?
>
> Learning to avoid losing critical information is essential for context management. Ideally, a well-trained MEM1 agent should be able to consolidate most critical information. In case some critical information is lost during the summarization, sometimes the agent will redo the search to retrieve back that piece of information. To better solve the problem, past context can be saved into an external memory store, allowing the MEM1 agent to retrieve it back when needed, possibly via some specialized tool calls.
>
> We wish to highlight that the MEM1 mechanism is orthogonal to the existing external memory mechanism (e.g., RAG, GraphRAG). We believe it is an interesting direction to explore the combination of the MEM1 rollout with explicit interaction and retrieval from an external memory for future work.

---

> ### Author Response · Authors · 2025-11-26
> **Response Part 4**
>
> > It could guide the model to use a summarization tool to condense all historical information into a fixed context window. What advantages does MEM1 offer compared to this method?
>
> We thank the reviewer for the interesting question. The scenario is analogous to the current popular reasoning mechanism. Instead of the popular approach of forcing the model to wrap its thinking in some XML tags such as <think></think>, one can, theoretically, choose to provide a "reasoning" tool for the model to call when it deems fit. However, in practice, the XML approach is still preferred over the tool call approach. We believe this is because **capabilities such as reasoning and context management are more fundamental capabilities that a model should master**. Therefore, it makes more sense to integrate them into the "chat template", training the model to master these skills, regardless of what tools are provided to it.
>
> One may ask whether it is possible to train reasoning or context consolidation using tool call format. While it is certainly possible, we believe doing so compromises the model's capability to correctly use other user-defined tools that may be passed in during runtime. This is because the agent that has been heavily trained to call the reasoning and context consolidation tools would often prefer these tools over user tools, even when it may be better to call the user tools. As such, the XML tag appraoch that separates fundamental capabilities from the flexible tool calling capability is adopted.
>
>
> > Do different search providers (such as Google Seach and Bing Search) have an impact on performance? What is the search parameter?
>
> We thank the reviewer for the question. Our experiments are primarily conducted using Google Search, where we set top-k value to be 3 (i.e., returning the top 3 search results). We have not experimented with other search engines. We believe that different search engines would certainly have some impact on performance, similar to how the choice of retriever affects the performance of RAG systems. However, since our method works well with both local RAG and Google Search as data sources, we do not think that additional evidence from other search engine providers is necessary to demonstrate the effectiveness of our approach. While the choice of search engine is indeed an important factor, it is somewhat beyond the scope of the current study, and we leave a deeper investigation of this aspect to future work.
>
>
> We hope that our response has sufficiently addressed the concerns and questions of the reviewer and improved the reviewer's opinion of our work. We would appreciate it if the reviewer would consider increasing the score if the reviewer finds our response satisfactory.

---

### Official Review · Reviewer_pi4p · 2025-10-27

**Soundness:** 2
**Presentation:** 3
**Contribution:** 2
**Rating:** 6
**Confidence:** 4

**Summary:**

This paper proposes MEM1, a reinforcement learning framework that jointly learns memory consolidation and reasoning for long-horizon LLM agents. Instead of storing full interaction histories or relying on external memory, MEM1 maintains a compact internal state updated at each step and discards the previous context. The framework is trained end2end via PPO using masked trajectories that enables stable and accurate policy optimization under MEM1’s memory-constrained execution. Experiments on multi-objective question answering and WebShop navigation demonstrate that MEM1 improves long-horizon task performance while significantly reducing inference memory and latency.

**Strengths:**

1. The paper proposes to treat memory as part of the policy and learn it jointly with reasoning, instead of relying on an external memory module or retrieval system. This is a refreshing take on long-horizon agents and feels conceptually meaningful: the model actually learns what to remember rather than being manually engineered to store past information.

2. Existing LLM agents don’t scale well over long contexts because their memory grows linearly (or worse) with interaction steps. MEM1 tackles this by enforcing a constant memory budget using an internal state. This is a practical and relevant contribution for real world agent deployment where inference cost matters.

3. The experimental analysis is relatively thorough and covers both quantitative and qualitative dimensions. On the quantitative side, the paper reports multiple evaluation metrics, including task accuracy, long-horizon generalization performance, peak memory usage, and inference efficiency. In addition, the paper goes beyond standard benchmark reporting by including behavioral analysis of the learned policy. The qualitative inspection of internal states and step-by-step trajectories provides useful insight into how MEM1 operates.

**Weaknesses:**

1. EM reward design lacks ablation study and maybe limits real world applicability . Although the paper adopts EM as the sole reward signal during RL training for QA tasks, it does not use ablation study to analyze or justify this specific reward choice. For example, they do not compare EM with other potential reward signals such as token-level F1, partial matching, or step-wise retrieval rewards that might better capture intermediate reasoning quality.  Also, the assumption of verifiable rewards such as EM restricts the scope of applicability. In most real world long-horizon tasks e.g., scientific literature review, legal case analysis, or customer support, ground-truth answers are ambiguous, subjective, thus EM-style rewards are unavailable.

2. Task Design lacks interdependent objectives. The multi-objective QA benchmark introduced in Section 3.3 constructs tasks by simply concatenating independent questions from HotpotQA and Natural Questions (e.g., “Which magazine was started first…?” and “The Oberoi family is part of a hotel company…?”). Crucially, these sub-questions share no entities, context, or logical dependencies, meaning the agent can solve them in isolation without cross-question reasoning or memory integration. As a result, the task primarily measures multi-task efficiency rather than true synergistic reasoning across interdependent objectives. One possible design can be referenced in M3Agent [1].

3. The evaluation lacks comparison to strong, lightweight memory compression baselines that do not require external modules or reinforcement learning. Notably absent is a simple LLM-based summarization baseline, where the same Qwen2.5-7B model summarizes the full history into a fixed-length context at each turn—a strategy that could achieve similar efficiency gains without RL. The “Truncation (prompt only)” ablation (Table 1) already achieves 0.396 EM on 16-objective QA—more than double the full-history Qwen2.5-7B-Instruct—suggesting that much of MEM1’s gain may stem from its prompt and rollout design rather than the RL policy itself.

[1] Seeing, Listening, Remembering, and Reasoning: A Multimodal Agent with Long-Term Memory

**Questions:**

1. Can the MEM1 framework be adapted to offline datasets (e.g., human or expert trajectories) and combined with RL algorithms like GRPO? What are the key challenges in doing so? For instance, would the unobserved internal state in offline data pose a fundamental identifiability issue?

2. The system prompt is excluded from token counting，but it’s part of the deployed model’s memory. Isn’t this misleading for real-world efficiency claims?

3. Training only on 2-objective tasks but testing up to 16, does performance gain come from generalization or simply from avoiding context collapse? Baseline models (e.g., Qwen2.5-14B) collapse on 16-objective tasks (Table 1: EM drops to near zero), likely because their context exceeds practical limits or attention dilution occurs. MEM1 avoids this by design. But is MEM1 truly reasoning better, or just not failing catastrophically?

---

> ### Author Response · Authors · 2025-11-26
> **Response Part 1**
>
> We thank the reviewer pi4p for taking time to provide the review and for complimenting that our approach is "a refreshing take on long-horizon agents and feels conceptually meaningful", our practical and relevant contribution, and that our experiments are relatively thorough. We address your concerns in the following.
>
> > EM reward design lacks ablation study and maybe limits real world applicability . Although the paper adopts EM as the sole reward signal during RL training for QA tasks, it does not use ablation study to analyze or justify this specific reward choice. For example, they do not compare EM with other potential reward signals such as token-level F1, partial matching, or step-wise retrieval rewards that might better capture intermediate reasoning quality.
>
> We thank the reviewer for suggesting the additional ablation. We follow popular existing approaches such as DeepSeek-R1 and Search-R1 that utilize verifiable matching of the final answer as the reward and did not consider its ablation in our paper due to limited content space. We believe it is certainly an interesting direction to consider different reward design in future to make our approach more scalable.
>
>
> > Also, the assumption of verifiable rewards such as EM restricts the scope of applicability. In most real world long-horizon tasks e.g., scientific literature review, legal case analysis, or customer support, ground-truth answers are ambiguous, subjective, thus EM-style rewards are unavailable.
>
> We thank the reviewer for suggesting the ablation. While it is a bit difficult to acquire these datasets and assess them, we conduct an ablation study using model-estimation reward instead of exact match reward to demonstrate the potential of MEM1 in training for these tasks. We experiment on WikiRAG and tabulate the results below. It can be observed that, although the trained model has low EM, it enjoys pretty good F1 score and model judgement score, suggesting the potential of applying MEM1 to train for tasks with non-verifiable rewards.
>
> | Setting | Number of Objectives | EM | F1 | Model Estimate Score | Peak Tokens |
> | --- | --- | --- | --- | --- | --- |
> | Model Estimate | 2 | 0.082 | 0.350 |0.726 | 620 |
> | Model Estimate | 8 | 0.14 | 0.918 | 1.65 | 627 |
> | Model Estimate | 16 | 0.20 | 1.59 | 3.75 | 700 |
>
>
> > Task Design lacks interdependent objectives. The multi-objective QA benchmark introduced in Section 3.3 constructs tasks by simply concatenating independent questions from HotpotQA and Natural Questions (e.g., “Which magazine was started first…?” and “The Oberoi family is part of a hotel company…?”). Crucially, these sub-questions share no entities, context, or logical dependencies, meaning the agent can solve them in isolation without cross-question reasoning or memory integration. As a result, the task primarily measures multi-task efficiency rather than true synergistic reasoning across interdependent objectives. One possible design can be referenced in M3Agent [1].
>
>
> We thank the reviewer for the great suggestions. We note that some existing works have attempted at some more intricate QA tasks such as browsecomp and webwalker. However, these tasks usually demand high reasoning capability of the base model which can be hardly learned purely from RL. As such, we used this concatenation task to highlight the strength of our proposed method.
>
> We are currently running an experiment to test whether MEM1 can be applied to DeepDive [1], a dataset that incorporates several dependent sub-problems. We follow a similar strategy as presented in the original paper to train a base 7B model with SFT. Different from them, we use the trajectories from our MEM1 rollout for SFT. Due to the considerable amount of compute required and engineering efforts to adapt our framework to this task, as well as the need to collect trajectories from large models, we are still training the model and the preliminary results show that our method remains to be effective for such more challenging tasks. We will update the results by the end of the rebuttal session. We appreciate the understanding of the reviewer for the delay.
>
> [1] Lu et al., DeepDive: Advancing Deep Search Agents with Knowledge Graphs and Multi-Turn RL, arXiv 2509.10446.

---

> > ### Author Response · Authors · 2025-11-26
> > **Response Part 2**
> >
> > > The evaluation lacks comparison to strong, lightweight memory compression baselines that do not require external modules or reinforcement learning. Notably absent is a simple LLM-based summarization baseline, where the same Qwen2.5-7B model summarizes the full history into a fixed-length context at each turn—a strategy that could achieve similar efficiency gains without RL. The “Truncation (prompt only)” ablation (Table 1) already achieves 0.396 EM on 16-objective QA—more than double the full-history Qwen2.5-7B-Instruct—suggesting that much of MEM1’s gain may stem from its prompt and rollout design rather than the RL policy itself.
> >
> >
> > We wish to clarify that the rollout mechanism is itself part of the contribution of this paper. Moreover, although the "truncate" baseline performs better than the collapsed Qwen2.5-7B-Instruct, it is still in no way close to the performance of MEM1, which highlights the significance of RL.
> >
> >
> > > Can the MEM1 framework be adapted to offline datasets (e.g., human or expert trajectories) and combined with RL algorithms like GRPO? What are the key challenges in doing so? For instance, would the unobserved internal state in offline data pose a fundamental identifiability issue?
> >
> > We thank the reviewer for the great question. We believe that MEM1 can be used in an offline setting. Specifically, we imagine that, for a decently large model, a "reinforced finetuning" approach can be used: Trajectories are first collected from running the model through the MEM1 rollout mechanism. Then, rejection sampling is conducted on the trajectories based on human expert labels, model estimate, or verifiable reward. The curated trajectories are then sent for SFT to produce a stronger model.
> >
> > Theoretically, RL algorithms other than PPO can also work. We choose PPO for training MEM1 as we think it provides a more stable training. As the base agent (Qwen2.5-7B in our paper) is not very good at following the MEM1 rollout mechanism at the start, it requires a quite large group size for GRPO to be efficient, which is beyond the capacity of our computing resources.
> >
> > We also wish to clarify that, in case there is a misunderstanding, that the internal state is full observable and interpretable, as it is presented as tokens generated by the model.
> >
> >
> > > The system prompt is excluded from token counting，but it’s part of the deployed model’s memory. Isn’t this misleading for real-world efficiency claims?
> >
> > We thank the reviewer for pointing out this important efficiency issue. First, some of our key efficiency metrics, such as inference time, include the system prompt for fair comparison (i.e., we use the original system prompts for baselines such as Search-R1 and DeepResearcher), and a short system prompt is indeed necessary for agentic reasoning tasks. The system prompt for MEM1 has a similar length (around 100 tokens, with MEM1 being slightly longer at around 200 tokens) to those used by other search-agent methods, and it is not significant compared to the number of tokens required for reasoning and task execution. In addition, system prompts in real-world deployments can leverage prefix caching for substantial efficiency gains, and memory usage is unlikely to pose a major issue when the agent processes multiple tasks of the same kind, as the cached prefix can be effectively reused.
> >
> >
> > > Training only on 2-objective tasks but testing up to 16, does performance gain come from generalization or simply from avoiding context collapse? Baseline models (e.g., Qwen2.5-14B) collapse on 16-objective tasks (Table 1: EM drops to near zero), likely because their context exceeds practical limits or attention dilution occurs. MEM1 avoids this by design. But is MEM1 truly reasoning better, or just not failing catastrophically?
> >
> > We thank the reviewer for raising the question. To see whether the performance gain comes from reasoning better, one could compare the performance between MEM1 and the two baselines with active context management: "Truncate" and "AMEM". "Truncate" uses the MEM1 rollout mechanism as it is and "AMEM" creates an agentic system with an external memory store. It can be observed that, at 16-objectives, both baselines perform better than Qwen2.5-14B, suggesting benefits from simply avoiding context collapse. However, the performance of MEM1 is still much higher than the two baselines, suggesting benefits from improved reasoning obtained by the MEM1 agent through RL.
> >
> >
> > We hope our responses have been satisfactory and improved your opinion. We would be grateful if you could increase your score if you found our responses have helped clarify your questions and concerns.

---

### Official Review · Reviewer_Dxgb · 2025-11-01

**Soundness:** 2
**Presentation:** 3
**Contribution:** 3
**Rating:** 6
**Confidence:** 4

**Summary:**

The paper introduces MEM1, an end-to-end RL framework for language agents that maintains constant memory over arbitrarily long multi-turn tasks by updating a compact shared internal state instead of appending full histories. Training integrates memory management with reasoning, and a scalable task-augmentation scheme builds multi-objective, multi-hop environments. On internal retrieval QA, open-domain web QA, and web shopping, MEM1 delivers strong—often SOTA—accuracy with markedly improved memory and inference efficiency, especially at long horizons.

**Strengths:**

- **Clear, Well-Structured Presentation**: The technical motivation, algorithm, and evaluation methodology are clearly described. Figures such as Figure 1 (“RL pipeline“) and Figure 2 ("performance and efficiency scaling") directly help crystallize the approach and the empirical insights.

- **Sound, End-to-End RL Optimization**: The use of reinforcement learning to train both reasoning and memory management jointly is well argued and empirically shown to benefit generalization to longer, more complex tasks. The masking-based trajectory construction and objective computation address the non-trivial technical challenge of dynamically changing agent context during rollouts.

**Weaknesses:**

- **Evaluation Reflects Synthetic Compositions, Not Open-Ended Dialogue.** Benchmarks are mainly constructed by composing QA subsets, which exercises multi-turn reasoning but biases toward compositional templates. Even WebShop, though interactive, is governed by predefined tasks, constrained action spaces, and scripted reward assumptions, so the current setup under-represents genuinely open-ended interactions with ambiguous goals or shifting task boundaries.


- **Insufficient Direct Comparison to Hierarchical and Structured Memory Architectures**: While the related work section covers many approaches, no empirical or conceptual comparison is offered with recent hierarchical/structured working memory proposals such as HiAgent [1], Zep[2] or [3]. These works explicitly address long-horizon memory management and could reveal functional or performance trade-offs.


 - **Lack of Ablations on Memory-Reasoning Coupling**: The core contribution is the tight integration of memory and reasoning, but no detailed ablation quantifies the benefit of this coupling versus traditional staged (e.g., memory-summarize-then-reason) approaches. Including such an ablation would clarify to what extent performance gains are due to this integration versus reinforcement learning or prompt design alone.

[1] Hu, Mengkang, et al. "Hiagent: Hierarchical working memory management for solving long-horizon agent tasks with large language model." _arXiv preprint arXiv:2408.09559_ (2024).

[2] Rasmussen, Preston, et al. "Zep: a temporal knowledge graph architecture for agent memory." _arXiv preprint arXiv:2501.13956_ (2025).

[3] Sun, Haoran, and Shaoning Zeng. "Hierarchical memory for high-efficiency long-term reasoning in llm agents." _arXiv preprint arXiv:2507.22925_ (2025).

**Questions:**

- What are the expected limitations or bottlenecks when scaling MEM1 to hundreds or thousands of objectives/turns? Are there inherent trade-offs between memory consolidation, retrieval accuracy, and forgetting?

- To what extent can MEM1’s method be positioned as an advance in lifelong or continual learning memory management (rather than purely RL agent efficiency) ?

- What specific performance gains (if any) are attributable to tightly integrating reasoning/working memory within a single update, versus staged or modular designs where reasoning and memory summarization are decoupled? A targeted ablation would aid understanding.

---

> ### Author Response · Authors · 2025-11-26
> **Response Part 1**
>
> We thank the reviewer Dxgb for taking time to provide the review and for commenting that our our presentation is clear and well structured and that our approach is sound. We address your concerns and questions below:
>
> > Insufficient Direct Comparison to Hierarchical and Structured Memory Architectures: While the related work section covers many approaches, no empirical or conceptual comparison is offered with recent hierarchical/structured working memory proposals such as HiAgent [1], Zep[2] or [3]. These works explicitly address long-horizon memory management and could reveal functional or performance trade-offs.
>
> We thank the reviewer for suggesting the additional baselines on hierarchical working memory. We will incorporate the suggested literature in our related work in our revised pdf. We also conduct an experiement with Zep (since it is the most popular on GitHub) on the WikiRAG benchmark and tabulate the results below. For Zep, we used the retrieved context (from Graphiti) as the internal state and apply the MEM1 rollout mechanism on a Qwen2.5-7B-Instruct model. It can be observed that, while Zep obtain decent performance when the number of objectives is low, it dose not scale well to higher number of objectives.
>
> | Method | no. objectives | EM | F1 | Peak Tokens | Time |
> | - | - | - | - | - | - |
> | (Graphiti) Zep | 2 | 0.362 | 0.456 | 445 | 2.55 |
> | (Graphiti) Zep | 8 | 0.067 | 0.074 | 610 | 2.60 |
> | (Graphiti) Zep | 16 | 0.042 | 0.049 | 654 | 2.81 |
> | MEM1 | 2 | 0.709 | 0.838 | 640 | 6.49 |
> | MEM1 | 8 | 1.87 | 2.31 | 810 | 8.68 |
> | MEM1 | 16 | 1.97 | 2.39 | 1040 | 8.70 |
>
> > Lack of Ablations on Memory-Reasoning Coupling: The core contribution is the tight integration of memory and reasoning, but no detailed ablation quantifies the benefit of this coupling versus traditional staged (e.g., memory-summarize-then-reason) approaches.
>
> We thank the reviewers for pointing out this important ablation study. We did consider separating memory and reasoning when we designed our approach, and our experiments showed that integrated memory is a more elegant approach that leads to higher Exact Match scores and fewer peak tokens. We train the agent with decoupled memory and reasoning by replacing the internal state with `<summary></summary><reasoning></reasoning>` tags and train the model with the same rollout and reinforcement learning approach. The full results on 2-objective, 8-objective, and 16-objective tasks for this ablation study are presented in the table below. Overall, **an integrated internal state results in better performance and reduced peak token count**. We will also update the manuscript to add this ablation study in the appendix.
>
> ### Ablation Study Results
>
> | Model                                      | 2-Objective EM | 2-Objective Peak (×10²) | 8-Objective EM | 8-Objective Peak (×10²) | 16-Objective EM | 16-Objective Peak (×10²) |
> |--------------------------------------------|----------------|--------------------------|----------------|--------------------------|------------------|---------------------------|
> | **MEM1-QA (integrated internal state)**     | 0.709          | 6.4 (0.02)               | 1.87           | 8.01 (0.06)              | 1.97             | 10.4 (0.09)               |
> | **MEM1-QA (separate memory + reasoning)**   | 0.686          | 11.8 (0.05)              | 1.78           | 9.63 (0.07)              | 1.92             | 13.37 (0.11)              |
>
>
> > What are the expected limitations or bottlenecks when scaling MEM1 to hundreds or thousands of objectives/turns? Are there inherent trade-offs between memory consolidation, retrieval accuracy, and forgetting?
>
> We thank the reviewer for the insightful question! We believe one of the possible hurdles could be to balance the compression frequency with the nature of task, current state of the task execution, etc.
>
> We agree with the reviewer that context management presents trade-offs between context length, decision making accuracy, and potentially forgetting past information due to context compression. We believe that this trade-off can be well exemplified by the delayed compression mechanism currently used by many long-horizon tasks. Instead of compressing context at every turn as MEM1 does currently, popular existing approaches compress memory intermittently based on some heuristics. Intuitively, compressing more aggressively leads to better consolidation but may compromise retrieval accuracy and information retention. On the other hand, compressing too sparsely risks having too long a context, leading to high inference cost and reduced model capacity for accurate retrieval. We thus believe it is an interesting future direction to incorporate learning the right time to consolidate context into the MEM1 framework to achieve the best trade-off between these conflicting objectives.

---

> ### Author Response · Authors · 2025-11-26
> **Response Part 2**
>
> > To what extent can MEM1’s method be positioned as an advance in lifelong or continual learning memory management (rather than purely RL agent efficiency) ?
>
> We thank the reviewer for the question. In fact, it is a question that the authors have put in quite some thought. For now, we consider MEM1 as primarily designed for long-horizon task solving rather than lifelong memory management. The main reason is that it is easier to organize information and memory **when one has a clear idea of the task at hand**. However, for lifelong conversation, the agent has to be prepared for unexpected queries, which makes it challenging to prune memory in advance, **as the agent cannot foresee what future queries to prepare for**.
>
> That said, under the assumption that the conversation is going to revolve around a fixed topic/goal, we believe that MEM1 can be utilized to implement a somewhat "infinite" context, achieving much better context understanding than the current standard and naive approach of simply discarding old context. Of course, we do not think that MEM1 is a replacement of external memory modules (e.g., RAG), but think that **it can be used in conjunction with them for enhanced performance in lifelong memory management**.
>
>
> > What specific performance gains (if any) are attributable to tightly integrating reasoning/working memory within a single update, versus staged or modular designs where reasoning and memory summarization are decoupled? A targeted ablation would aid understanding.
>
> Please see our response and experiment details above.
>
> We hope our responses have been satisfactory and improved your opinion. We would be grateful if you could increase your score if you found our responses have helped clarify your questions and concerns.

---

### Official Review · Reviewer_zPnP · 2025-11-02

**Soundness:** 3
**Presentation:** 3
**Contribution:** 4
**Rating:** 8
**Confidence:** 4

**Summary:**

This paper introduces MEM1, a reinforcement learning (RL) framework that trains LLM agents to handle long-horizon tasks by managing its own contextual memory. Instead of suffering from context exploding in the conventional approaches that append all history in the context, MEM1 trains, via RL, an agent to autonomously and iteratively consolidate its existing context, and learn to synergise its reasoning and information compression. The paper conducts extensive and delivers promising experimental results, where a 7B model trained with MEM1 outperforms a much larger 14B model as well as existing approaches like Search-R1 with significantly improved performance and memory usage on 16-objective tasks. The paper also demonstrates generalizability of the method on WebShop task.

**Strengths:**

- The problem is well motivated, targeting a critical bottleneck of using LLM agents for real-world problems. Curbing the unbounded context growth finds many applications in AI agent applications, including deep research, web agents, game playing agents, etc..
- The proposed method is simple but effective. MEM1 encourages the AI agent to synergise reasoning and memory consolidation by designing a clever rollout mechanism in an RL pipeline. The simplicity and end-to-end nature also implies the potential scalability to solving highly complicated long-horizon tasks.
- The paper demonstrates impressive empirical results, where MEM1 reliably outperforms baseline methods as the tasks become more long-horizon by increasing the number of objectives.

**Weaknesses:**

- MEM1’s reliance on verifiable and dense reward signals limits its applicability to open-ended or subjective tasks. Many realistic LLM-agent settings (e.g., creative reasoning and open-ended QA) lack such clear supervision. It is interesting how it can be extended to cases where the rewards are more implicit.
- Some presentation issues: (1)The naming is a bit messy. The paper has used “long-turn”, “long-horizon”, and “multi-turn” throughout. Do they mean the same thing? If so, the authors should standardize the term. (2) The paper repeatedly sells “constant context size,” but the evidence is proxy metrics (peak token count excluding system prompt) that still increase with objectives (e.g., 6.40 -> 10.4×100 tokens from 2->16 objectives). There is no formal upper bound on internal state (IS) length or a proof that memory is constant w.r.t. horizon. I suggest the authors restate claims as “near-constant peak tokens under our rollout policy” to make it more rigorous.

**Questions:**

- What are some of the failure cases when MEM1’s context consolidation fails to be effective?
- On Figure 2, why is the peak token for 8x objectives more than 16x objectives for some models?

**Details Of Ethics Concerns:**

I don't have any ethics concerns.

---

> ### Author Response · Authors · 2025-11-26
> **Response Part 1**
>
> We thank the reviewer zPnP for taking time to provide a thoughtful review and for complimenting the motivation, effeciveness, and the contribution made by this paper. We address the reviewer's concerns and questions in the following.
>
> > MEM1’s reliance on verifiable and dense reward signals limits its applicability to open-ended or subjective tasks. Many realistic LLM-agent settings (e.g., creative reasoning and open-ended QA) lack such clear supervision. It is interesting how it can be extended to cases where the rewards are more implicit.
>
> We thank the reviewer for suggesting the setting with non-verifiable or implicit reward. In our current work, we focused on problems with verifiable rewards to enable quantitative assessment of MEM1's performance gain. To demonstrate how MEM1 can potentially be applied in scenarios with non-verifiable reward, we conducted an extra experiment. In this setting, we train the MEM1 agent with **model-estimate reward** using one the current frontier models as a judge to assign scores, instead of exact match or F1 score reward. The model-estimate reward mimics situations where there is no golden answer. The results are tabulated below. While the model-estimate reward achieves low exact-match because model estimate reward doesn't enforce formatting and the trained agent tends to answer with complete sentences rather than exact phrases, it obatins very high estimate score and decent F1 score, suggesting that the model can learn from non-verifiable reward.
>
> | Setting | Number of Objectives | EM | F1 | Model Estimate Score | Peak Tokens |
> | --- | --- | --- | --- | --- | --- |
> | Model Estimate | 2 | 0.082 | 0.350 |0.726 | 620 |
> | Model Estimate | 8 | 0.14 | 0.918 | 1.65 | 627 |
> | Model Estimate | 16 | 0.20 | 1.59 | 3.75 | 700 |
>
>
>
> > (1)The naming is a bit messy. The paper has used “long-turn”, “long-horizon”, and “multi-turn” throughout. Do they mean the same thing? If so, the authors should standardize the term.
>
> We thank the reviewer for pointing out the presentation issue. Yes, all these three terms refer to the same concept. While the terminology has not been very standardized by the time of writing this paper, we have noticed that the emerging consensus in the community is "long-horizon". We will standardize all terms to "long-horizon" and upload a revised pdf version.
>
> > The paper repeatedly sells “constant context size,” but the evidence is proxy metrics (peak token count excluding system prompt) that still increase with objectives (e.g., 6.40 -> 10.4×100 tokens from 2->16 objectives). There is no formal upper bound on internal state (IS) length or a proof that memory is constant w.r.t. horizon. I suggest the authors restate claims as “near-constant peak tokens under our rollout policy” to make it more rigorous.
>
> We thank the reviewer for suggesting the change in phrasing for more rigorous statements. During our experiments, we do not explicitly limit the length of the internal state and therefore there is indeed no guarantee of the length of the internal state. In our experiments, we found that across iterations, the model’s internal state does not grow drastically because it naturally learns to discard information that is no longer useful (see more details in our response to reviewer CDp4). That being said, the context size results from empirical practice with no rigorous guarantee. We agree that “near-constant peak tokens” is a more rigorous term. We will revise the phrasing as you suggested for a clearer and more rigorous presentation.

---

> > ### Author Response · Authors · 2025-11-26
> > **Response Part 2**
> >
> > > What are some of the failure cases when MEM1’s context consolidation fails to be effective?
> >
> > We thank reviewers for asking about the failure cases, and we believe a comprehensive analysis of failure cases can give us more insights into the limitations of the proposed method and offer insights for future improvements. After careful examination of the failed trajectories from our 2-question task, **we identify 5 representative ways that the model failed to answer all questions correctly**. Overall, 3 out of these 5 failure modes are related to general failures in the search agent, and 2 are directly related to our iterative compression rollout and compositional tasks in specific. We name them entity "ambiguity", "hallucination instead of search", "logical fallacies", "insufficient query refinement", and "overwriting correct answers due to loss of the original reasoning path". In the following table, we provide full explanations for these failure cases and an example illustrating each failure case.
> >
> > | Failure Mode | Explanation | Example | Statistics |
> > | --- | --- | --- | --- |
> > | Entity Ambiguity | The agent chooses the wrong real-world entity among several with the same or similar name, then builds the entire answer on that wrong entity. | The query asks for the birthplace of the performer of *“Step Into The Light (Myra Song)”*. The search results provide two different entities sharing the name “Myra”: (1) *Myra*, the Mexican-American pop singer who actually performs the song, and (2) *Myra Taylor*, an unrelated American jazz singer whose biography conveniently lists a birthplace. The model chooses the wrong entity—Myra Taylor—ignoring the explicit evidence in the first document that identifies the correct performer. It then builds its final answer on this mistaken identity, reporting *“Bonner Springs, Kansas”*, which is correct for Myra Taylor but irrelevant to the song. | **164 (20.1%)** |
> > | Hallucination Instead of Search | The agent makes up facts (dates, places, relationships, biographies, etc.) instead of using search or acknowledging uncertainty. | When asked about the occupations of *Joni Jenkins* and *Martin Shapiro*—two names for which the model has no reliable prior knowledge—it fully fabricated specific occupations: *“Joni Jenkins’s occupation is Interior Designer. Martin Shapiro’s occupation is Lawyer. I recalled that she is known for her work as an interior designer… Since I have the required information without needing to search, I can provide the answers directly.”* It then output: *“\<answer> Interior Designer; Lawyer \</answer>”*. Both answers are wrong (gold: *politician*; *screenwriter*). | **391 (47.9%)** |
> > | Logical Fallacies | The agent uses invalid or irrelevant criteria—such as spurious proxies or unfounded assumptions—to compare multiple entities and draw conclusions, resulting in logically incorrect compositional reasoning. | The question is explicitly about which director is older, so it requires comparing two people’s ages. Instead of retrieving or reasoning about the directors’ birth dates, the agent looks at the films’ release years (2005 vs 2008), assumes the directors are “of similar ages when they directed these films,” and concludes that the director of the earlier film must be older. A snippet from the internal reasoning: *“After researching, I found that Flying Boys was released in 2005, and Dragged Across Concrete was released in 2008. Assuming both directors were of similar ages when they directed these films, the director of Flying Boys is older since it was released earlier.”* This misuse of an irrelevant proxy leads to a logically invalid conclusion. | **31 (3.8%)** |
> > | Insufficient Query Refinement | The agent repeats nearly identical, low-yield queries and never significantly refines its search strategy after getting irrelevant or partial results, so it fails to retrieve the crucial information (Insufficient Query Refinement After Irrelevant Results and Early Termination). | The task asks, “In what country is Awe?; In what country is Mrákotín?”. For **Awe**, the agent repeats the identical low-yield query `<search> Location of Awe</search>` three times (t0, t2, t4), continually retrieving pages about the *emotion* “awe” or the *Bridge/River Awe* in Scotland—none of which answer the geographic question—yet it never refines the query with clarifiers such as “town,” “place name,” or “USA,” and stays stuck. For **Mrákotín**, it similarly repeats `<search> Location of Mrákotín</search>` (t1, t3), again receiving irrelevant results (e.g., *Mrkonjić Grad*) and failing to adjust its search strategy. Only on the final forced turn does it accidentally retrieve the correct village entry. The failure comes from not adapting the search strategy. | **35 (4.3%)** |

---

> > > ### Author Response · Authors · 2025-11-26
> > > **Response Part 3**
> > >
> > > Contd.
> > > | Failure Mode | Explanation | Example | Statistics |
> > > | --- | --- | --- | --- |
> > > | Overwriting Correct Answers Due to Loss of the Original Reasoning Path | In compositional settings, the agent remembers what answer it got but not why or how, so when new partial or conflicting information appears, it overwrites or corrupts the original answer instead of defending it using the original reasoning chain (discarding the step-by-step path after storing an answer). | For the query *“According to Greek legend, King Midas was blessed with the gift that all he touched would turn to what?; What type of rain did Prince sing about in 1984?”* the agent initially retrieves the correct evidence that Prince’s 1984 song is *“Purple Rain”* and thus has everything needed to answer the “type of rain” question. However, later reasoning goes astray. Because the agent does not retain or re-access the full history of how its earlier (correct) conclusion was derived, it becomes overly focused on the song’s genre description (“power ballad”) and incorrectly asserts that there is “no specific type of rain” mentioned. It overwrites the correct candidate “Purple Rain” with the wrong answer “power ballad,” showing how losing the detailed reasoning path makes the model easy to mislead. | **7 (0.9%)** |
> > > | Others | Non-typical or miscellaneous failures that are not captured by the five main failure modes above (e.g., missing information in the corpus, formatting issues, or idiosyncratic edge cases). |  | **188 (23.4%)** |
> > >
> > >
> > > There are some non-typical failures that are not captured under these 5 failure modes, and we categorize them as "others". We found that some failures are hard to avoid because the Wikipedia corpus we used as our information source may not contain answers for all the questions, and some failures may be due to formatting issues. We also calculate the percentage of each type of failure among all trajectories by labeling the error type of each trajectory. Among the 1000 trajectories we examined, where the model answered one question correctly for 397 tasks and both questions correctly for 156 tasks, the percentages and failure counts are reported in the table below as well. Please note that one failed trajectory might contain more than one failure mode, so the sum might not match the number of failed trajectories exactly, but this should give a good reference for the failure mode distribution.
> > >
> > > > On Figure 2, why is the peak token for 8x objectives more than 16x objectives for some models?
> > >
> > > Thank you for the great question. The reason that some models have lower peak tokens for 16x objectives is that these models have **"collapsed" during the rollout** for solving a complex questions. We notice that, especially for base models not trained on long-horizon agentic reasoning tasks before, such as Qwen2.5-7/14B-Instruct, it can **give up half-way for the more demanding 16-objective questions**: Instead of trying to solve as many problems as possible, these models quickly return an answer after only a few search rounds, resulting in a reduced peak token count.
> > >
> > > We would like to once again thank you for your positive review. We hope our clarifications have been sufficient in addressing your concerns and questions.

---

> > > > ### Comment · Reviewer_zPnP · 2025-11-27
> > > > **Response to Rebuttal**
> > > >
> > > > Thanks for the detailed rebuttal! I do not have any other questions at this time.

---

### Author Response · Authors · 2025-12-03
**Summary of Rebuttal Discussion**

Dear Area Chair,

To facilitate the rebuttal process, we have consolidated below a concise summary of the main concerns from the reviewers and how we addressed them.

**1. Application of MEM1 for Open-domain Tasks (Reviewer zPnP, Dxgb, and pi4p)**
- Concern: The experiments conducted in the paper use only exact mactch score as the reward. The reviewers wonder whether the method can be generalized to open-domain tasks with no such easily verifiable scores.
- Response: We have supplemented the experiments where we train MEM1 using model-estimated score instead of exact match. Evaluation shows that the trained MEM1 agent, while achieving much lower EM score, attains very high model estimated score and F1 score, suggesting that the trained agent is able to capture the essence of the answer. A table of the experiment result can be found in our response to the two reviewers.

**2. Lack of Failure Case Analysis (Reviewer zPnP and CDp4)**
- Concern: While the paper has presented some qualitative analysis of the success cases, there lacks analysis of failure cases.
- Response: We have provided a table detailing a few categories of failure cases with examples and statistical distributions. The table can be found in our response to the reviewer zPnP.

**3. Question over the length of the internal state mechanism (Reviewer zPnP and CDp4)**
- Concern: The reviewers raised questions about whether the internal state can maintain a "hard" constant length or that the length of the internal state increases with the number of interaction turns with the environment.
- Response: We have clarified in our responses that we do not explicitly limit the size of the internal state. The constant length is an observation. We have revised the phrasing in our paper to "near-constant" to be more rigorous. We have also provided an empirical analysis of the size of the internal state in our response to reviewer CDp4, showing that the length of the internal state does not increase over the number of interaction turns with the environment, although the length does increase slightly as the task becomes more difficult.

**4. More difficult benchmark where searches are dependent. (Reviewer Dxgb and CDp4)**
- Concern: The task considered by the paper does not reflect the more real-life scenarios where each query is broken into interdependent sub-queries instead of the independent sub-queries discussed in the paper.
- Response: We have conducted experiments that train MEM1 agent using DeepDive [1] which composes multiple inter-related sub-queries to form composite queries. We first SFT a Qwen2.5-7B model using MEM1 trajectories obtained from a frontier model. Then, we perform RL on the SFT model. We report the evaluation in the table below. For all datasets, we randomly take 100 samples for evaluation. It can be observed that MEM1 agent can also achieve performance improvement on more difficult tasks while achieving much lower peak token count compared to baseline methods. Notably, compared to the SFT agent, the RL agent obtains much reduced peak tokens while achieving higher accuracy.


| agent/model | dataset | accuracy (model estimated score) | peak token |
|---|---|---|---|
|Qwen2.5-7B-Instruct| DeepDive Test Set | 6.06% | 4900 |
| MEM1 SFT | DeepDive Test Set | 11.1% | 2125 |
| MEM1 RL | DeepDive Test Set | **14.0%** | **914** |
|Qwen2.5-7B-Instruct (with MEM1 Rollout) | Browsecomp | 0% | 1064 |
|Qwen2.5-7B-Instruct| Browsecomp | 4.0% | 7740 |
| MEM1 SFT | Browsecomp | 4.0% | 2222 |
| MEM1 RL | Browsecomp | **5.0%** | **1124** |


Additionally, reviewer Dxgb has suggested some additional baseline comparisons and ablation studies. We have provided the results in our response to the reviewer which demonstrate the effectiveness of MEM1.

We sincerely thank all the reviewers for their constructive feedback and thoughtful comments. The suggestions and comments have been a great contribution to the improvement of our work.

[1] Lu et al., DeepDive: Advancing Deep Search Agents with Knowledge Graphs and Multi-Turn RL, arXiv 2509.10446.

---

### Meta-Review · Area_Chair_XG8A · 2026-01-08

**Summary:**

This submission introduces MEM1, a reinforcement learning-based framework designed to enable language agents to solve long-horizon, multi-turn tasks with constant context size. By updating a shared internal state at each turn, MEM1 consolidates memory and supports reasoning, discarding irrelevant or redundant information. Extensive experiments demonstrate that MEM1 owns the potential of reasoning-driven memory consolidation as an efficient and scalable solution for multi-turn task-solving agents.

**Reviewer Concerns:**

Generally, the reviewers' concerns can be summarized as the following points:

1. Application of MEM1 for Open-domain Tasks (Reviewer zPnP, Dxgb, and pi4p). The experiments conducted in the paper use only exact mactch score as the reward. The reviewers wonder whether the method can be generalized to open-domain tasks with no such easily verifiable scores.
2. Lack of Failure Case Analysis (Reviewer zPnP and CDp4). While the paper has presented some qualitative analysis of the success cases, there lacks analysis of failure cases.
3. Question over the length of the internal state mechanism (Reviewer zPnP and CDp4). The reviewers raised questions about whether the internal state can maintain a "hard" constant length or that the length of the internal state increases with the number of interaction turns with the environment.
4. More difficult benchmark where searches are dependent. (Reviewer Dxgb and CDp4). The task considered by the paper does not reflect the more real-life scenarios where each query is broken into interdependent sub-queries instead of the independent sub-queries discussed in the paper.

**Reviewer Scores:**

Four reviewers respectively rated the submission with the scores 8, 6, 6, 6. According to the reviewers' feedback, the main concerns focus on the insufficient comparison and analysis, and after the rebuttal, the authors provided more complete demonstration in these directions. After reading the rebuttal and the revision, AC considered that four reviewers may not completely increase scores but at least hold the positive attitude regarding the submission, which tends to be accepted.

---

### Decision · Program_Chairs · 2026-01-26

Accept (Poster)